# Isoform- and ligand-specific modulation of the adhesion GPCR ADGRL3/Latrophilin3 by a synthetic binder

Szymon P. Kordon [1,2,3], Przemysław Dutka[1,4], Justyna M. Adamska [1,2,3], Sumit J. Bandekar [1,2,3], Katherine Leon[1,2,3], Satchal K. Erramilli[1], Brock Adams[1,2,3], Jingxian Li[1,2,3], Anthony A. Kossiakoff [1,3] & Demet Araç [1,2,3] ✉

Adhesion G protein-coupled receptors (aGPCRs) are cell-surface proteins with large extracellular regions that bind to multiple ligands to regulate key biological functions including neurodevelopment and organogenesis. Modulating a single function of a specific aGPCR isoform while affecting no other function and no other receptor is not trivial. Here, we engineered an antibody, termed LK30, that binds to the extracellular region of the aGPCR ADGRL3, and specifically acts as an agonist for ADGRL3 but not for its isoform, ADGRL1. The LK30/ADGRL3 complex structure revealed that the LK30 binding site on ADGRL3 overlaps with the binding site for an ADGRL3 ligand – teneurin. In cellular-adhesion assays, LK30 specifically broke the trans-cellular interaction of ADGRL3 with teneurin, but not with another ADGRL3 ligand – FLRT3. Our work provides proof of concept for the modulation of isoform- and ligand-specific aGPCR functions using unique tools, and thus establishes a foundation for the development of fine-tuned aGPCR-targeted therapeutics.

With 33 members in the human genome, adhesion G protein-coupled receptors (aGPCRs) represent the second-largest subfamily of GPCRs. Genetic studies have identified critical roles for aGPCRs in development, immunity, and neurobiology[1–7] linking them to numerous diseases including neurodevelopmental disorders, deafness, male infertility, schizophrenia, immune disorders, and cancers[2,8–15]. While aGPCRs are crucial surface receptors involved in numerous physiological processes[16], establishing an understanding of their mechanisms of action at the molecular level remains an ongoing challenge, as has been the development of tools to more effectively study them and to modulate their functions.

In addition to their signaling seven transmembrane (7TM) helices that exist in all GPCRs, a hallmark of aGPCRs is their large multidomain extracellular regions (ECRs)[17–19]. The multidomain ECRs of aGPCRs can bind to protein ligands that are either on neighboring cells or in the extracellular matrix, effectively regulating receptor function and downstream signaling[11,20–27]. The ECRs of aGPCRs range from ~200 to

~5900 amino acids and can be comprised of combinations of approximately 30 different types of adhesion domains that include the GPCR Autoproteolysis Inducing (GAIN), Epidermal Growth Factor (EGF), lectin (Lec), immunoglobulin (Ig), cadherin, olfactomedin (Olf), pentraxin, laminin and other domains. While the GAIN domain is conserved in aGPCRs, other extracellular domains vary between aGPCRs and enable each receptor to bind specifically to unique ligands and thus mediate different biological functions.

Several studies, including ours, have identified ECR-targeted synthetic proteins that activate or inhibit aGPCRs[22,28–30]. However, none of these studies addressed specificity against different aGPCR isoforms or provided insight into strategies for modulating particular aGPCR-ligand interactions. Often therapeutically active reagents that target cell-surface receptors have problems with specificity because they also act on unintended receptors such as other isoforms[31,32]. For these reasons, further work for understanding how to specifically target aGPCRs and fine-tune aGPCR functions is needed. As most

[1]Department of Biochemistry and Molecular Biology, The University of Chicago, Chicago, IL, USA. [2]Neuroscience Institute, The University of Chicago, Chicago, IL, USA. [3]Institute for Biophysical Dynamics, University of Chicago, Chicago, IL, USA. [4]Present address: Division of Chemistry and Chemical Engineering and Division of Biology and Biological Engineering, California Institute of Technology, Pasadena, CA, USA. ✉e-mail: arac@uchicago.edu

aGPCRs mediate multiple functions, targeting only a single function of the receptor, while leaving the other functions unaffected can be challenging. The supposition of our work is that antibody-like molecules that target the ECRs of aGPCRs may result in highly specific functional modulators because aGPCR ECRs are much more diverse than their 7TM regions[33]. This approach can also regulate specific aGPCR-related activities by targeting the specific aGPCR-ligand interaction that is responsible for the particular activity. In addition, aGPCRs have been shown to have some roles that depend only on their ECR, making them independent of their TM region[34–36]. In such cases, ECR-targeted reagents are the only way to modulate these receptor-induced activities.

Latrophilins (ADGRLs, LPHNs) constitute a model aGPCR subfamily that play crucial roles in embryogenesis, tissue polarity, and synapse formation; their mutations are associated with numerous cancers and attention deficit-hyperactivity disorder (ADHD)[37–44]. There are three ADGRL isoforms (paralogs) in vertebrates (ADGRL1-3). ADGRL1 and ADGRL3 are primarily expressed in the brain, whereas ADGRL2 is ubiquitously expressed[45,46]. The distribution of the ADGRL isoforms in different tissues suggests that each may contribute to a different set of diverse processes. Studies in rats showed that ADGRL3 knockout (KO) results in hyperactivity and increased acoustic reactivity; protein levels of ADGRL1 and ADGRL2 isoforms remained unaltered, showing no compensatory upregulation in ADGRL3 KO rats[47]. A recent study in mice showed that the ADGRL2 and ADGRL3 isoforms mediate the formation of distinct synapses on the same hippocampal neuron and cannot compensate for each other suggesting each isoform has distinct functions, which helps explain the specificity of synaptic connections[5,40,48]. Furthermore, ADGRL isoform-specific remodeling of the actin-associated complexes in HEK293T cells was reported in response to teneurin binding[49]. Thus, affinity reagents that target inhibition of isoform-specific interactions are desirable as they will be able to modulate different ADGRL functions.

ADGRL ECRs are comprised of a Lectin (Lec) domain, an Olfactomedin (Olf) domain, a stalk-like region, followed by a Hormone Binding (HormR) domain and a GAIN domain which directly precedes the seven-transmembrane helix region (7TM) (Fig. 1a)[19,20]. ADGRLs have numerous endogenous ligands and likely have more unknown ones that will be revealed with further study. Most ligands interact with all three ADGRL isoforms; however, others bind preferably to one of the isoforms[50]. Some of the most highly studied ADGRL ligands include teneurins (TENs)[38,42,43,51] and fibronectin leucine-rich repeat transmembrane proteins (FLRTs)[23,52], although other less studied interactions with neurexins (NRXs)[53] and contactins[54] have also been reported. ADGRL interactions with both TENs and FLRTs have been shown to induce excitatory synapse formation and specification[42,43]. The excitatory synapse formation depends on both TEN and FLRT interactions with ADGRL, although each ligand might be important for different functions. Thus, molecules that can inhibit the interactions between ADGRL isoforms and their ligands are desirable to regulate ADGRL functions more specifically. Currently, no synthetic binders have been reported for any ADGRL isoforms.

In this paper, we used ADGRLs as an exemplary aGPCR system to demonstrate that the ECRs of aGPCRs can be targeted by synthetic binders in a ligand- and isoform-specific manner. Employing phage-display technology, we have generated and characterized a synthetic antibody fragment (sAB) against the ADGRL3 ECR that targets a single domain of ADGRL3. Cell-based signaling assays showed that this sAB acts as an agonist for ADGRL3, but not for ADGRL1, although it binds to both with similar affinities. The crystal structure of the sAB in complex with the Lec domain of ADGRL3 showed that the sAB overlapped with the TEN2 binding epitope on ADGRL3. Herein, we show that specifically breaking ADGRL3's interaction with one ligand—TEN2—still allows maintaining the interaction with another—FLRT3. In this work, we have developed valuable tools that will enable further studies of ADGRL function and provide the principles for fine-tuned modulation of aGPCR signaling and downstream biological function.

## Results

### A high-affinity domain-specific antibody directed to the extracellular region of ADGRL3

In order to generate high-affinity sABs against ADGRL3, biotinylated full-length ECR of ADGRL3 was subjected to phage display selection using a high-diversity synthetic phage library based on a humanized antibody Fab scaffold[55]. To increase the specificity and affinity of the sABs, four rounds of selection were performed, with decreasing concentrations of target in each round (Supplementary Fig. 1a). After selection, 96 binders were screened using a single-point phage enzyme-linked immunosorbent assay (ELISA) (Supplementary Fig. 1b). The clones showing high ELISA signal intensity, when compared to control wells were sequenced, identifying a total number of 10 unique binders against ECR of ADGRL3. Selected phagemids were then cloned into sAB protein format, expressed in *E. coli* and purified by protein L affinity chromatography for further characterization (Supplementary Fig. 1c). These antibodies are the first that target the ADGRL subfamily of aGPCRs.

To determine the epitope of the selected sABs on ECR of the ADGRLs, we performed a single-point protein ELISA, utilizing fragments of either human ADGRL3 or rat ADGRL1 (Fig. 1b). Epitope mapping experiments revealed that three out of six ADGRL3 sABs (LK29, LK30 and LK31) bound to the N-terminal Lec domain of the receptor, while sAB LK12 recognized only the construct including both Lec and Olf domains. Interestingly, we also observed that all of the ADGRL3 Lec domain binders can also recognize the Lec domain from ADGRL1.

We determined the binding kinetics of the sABs to their targets by surface plasmon resonance (SPR) (Fig. 1c-e, Supplementary Fig. 1d, e, Supplementary Fig. 2). The binding constants ($K_d$) for all of these sABs are in low nanomolar (nM) range with most being characterized by slow dissociation rates ($k_{off} < 10^{-3}\,s^{-1}$) (Supplementary Fig. 1f). From the most promising cohort, we focused on the best expressing sAB, LK30, which binds to the Lec domain of ADGRL3 with 7 nM affinity and Lec domain of ADGRL1 with 11 nM affinity for future experiments (Supplementary Fig. 1g).

Binding of LK30 to the ECR fragments (Lec and Lec-Olf) of ADGRL3 in solution was confirmed by size exclusion chromatography (SEC). There was a shift in the retention volume of the ADGRL3/LK30 complexes compared with the purified ADGRL3 protein fragments alone and we observed co-elution of both proteins, as analyzed by SDS-PAGE (Fig. 1f, Supplementary Fig. 1h). In order to test whether the sABs that were selected against the ECR of ADGRL3 can also bind to full-length ADGRLs expressed on the cell surface, we utilized flow cytometry experiments (Fig. 1f, Supplementary Fig. 3b). We expressed either full-length ADGRL3 or ADGRL1 constructs in HEK293T cells and added increasing concentrations of LK30. To detect LK30 binding, we utilized a fluorescently labeled anti-human IgG sAB-fragment specific antibody. For each LK30 concentration, the mean fluorescent intensity (MFI) was measured and these values were plotted against the concentration of LK30 to estimate an apparent affinity. We determined that LK30 binds to the full length ADGRL3 and ADGRL1 expressed on the cell surface-expressed receptors with high affinity (131 nM and 147 nM, respectively), indicating that its binding epitope is not hindered by the proximity of the membrane or from possible differences in glycosylation patterns in mammalian cells.

### LK30 specifically modulates downstream signaling of ADGRL3, but not ADGRL1

Previous work had shown that the binding of biological ligands to the ECR of other GPCRs can alter receptor signaling ability[22,28–30].

ADGRL3 signals through $G\alpha_{12/13}$, $G\alpha_i$ or $G\alpha_q$ proteins, with $G\alpha_{12/13}$ being activated the most[44,56]. We have previously reported that ADGRL3 and ADGRL1 are active in a serum response element (SRE)-luciferase assay, that can detect, among others, activation of $G_{12/13}$, which is upstream of RhoA and SRE[44]. Therefore, we aimed to test the effect of sAB binding on ADGRL3 basal activity, using an SRE-luciferase assay which allows for quantification of a stable luminescent signal from firefly luciferase that is correlated with the receptor activity. LK30 treatment of ADGRL3 transfected cells resulted in increased basal signaling of the receptor, as seen in the SRE assay results (Fig. 2a). The LK30 effect was specific to ADGRL3, as cells transfected with empty vector did not show any significant change in signaling. Treatment with 1 μM of LK30

increased ADGRL3 basal signaling with an $EC_{50}$ of 42 nM. This was an approximately 2.5-fold increase in signaling compared to the basal activity of the receptor (Fig. 2b). Interestingly, we observed no effect of LK30 addition on ADGRL1, even at higher LK30 concentrations (up to 10 μM), despite its ability to bind to ADGRL1 (Fig. 2a, b). Thus, these results provide evidence for isoform-specific modulation of ADGRLs by the synthetic ligand LK30.

Previously, we have reported basal activity of ADGRLs in the luciferase-based GloSensor assay (Promega), which reports increase or decrease of intracellular cAMP levels in mammalian cells[26,44]. Using a modified cAMP-assay, where we elevate the cAMP levels by co-transfection of ADGRLs with the β2-adrenergic receptor (ADRB2) and

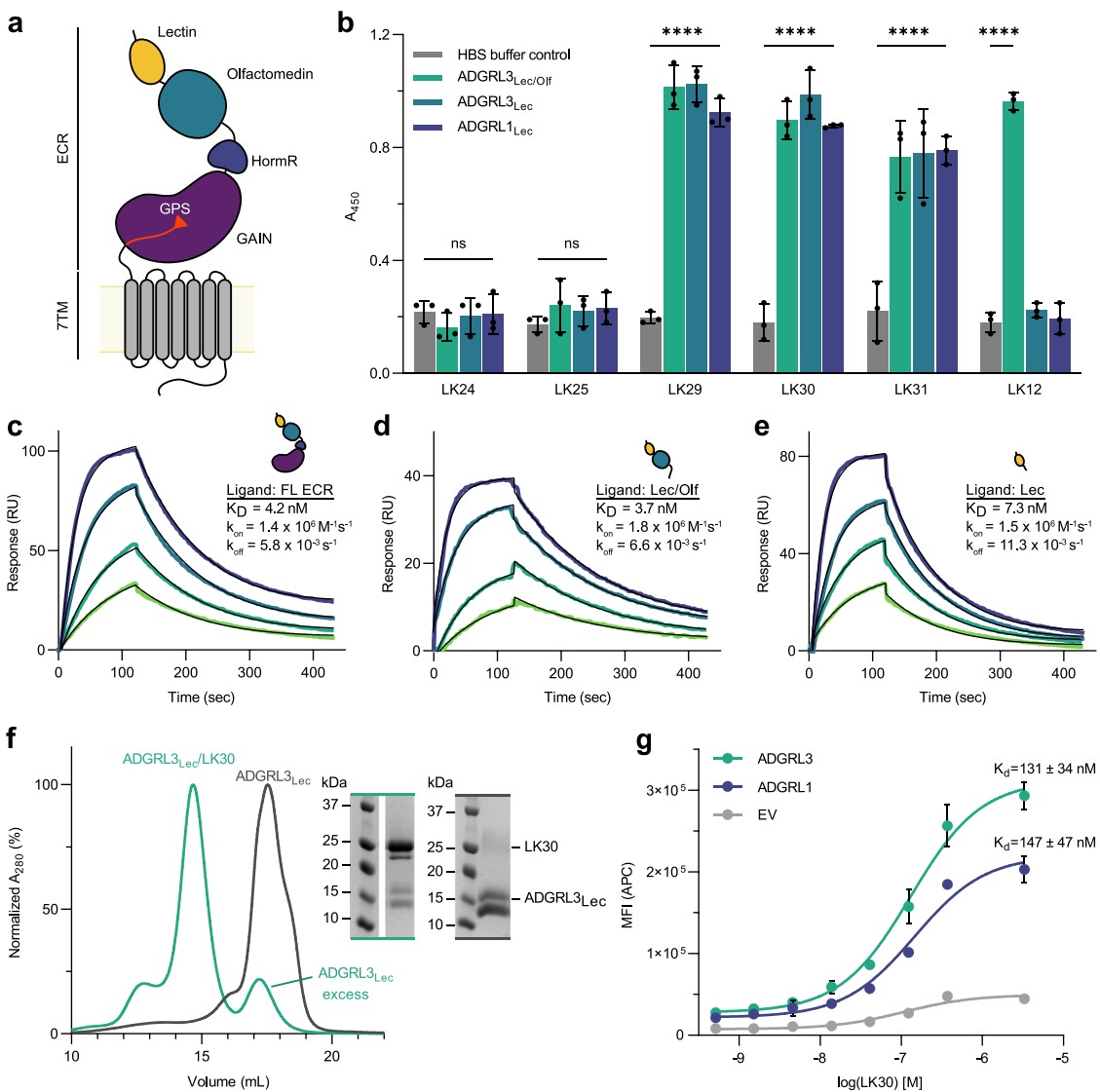

Fig. 1 | Characterization of the sABs against ADGRL3. a Schematic diagram of full-length human ADGRL3. The Lectin domain is colored yellow, Olfactomedin - cyan, Hormone Binding Region - navy, GAIN - purple and 7TM - gray. The last β-strand of the GAIN domain is colored red and the autoproteolysis site within the GAIN domain (GPS) is presented as a triangle. b Representative single-point protein ELISA of the antibody binders obtained from phage display. Epitope mapping shows that sABs LK29-31 bind to the lectin domain of both ADGRL3 and ADGRL1. Data are presented as mean ± SD of three repeats ($n = 3$), $^{ns}p = 0.9988$; $^{ns}p = 0.6364$; $^{****}p < 0.0001$ vs. HBS buffer treatment; two-way ANOVA. Source data are provided as a Source Data file. c–e Surface plasmon resonance measurements of the LK30 binding to (c) purified ADGRL3 ECR, (d) Lec/Olf fragment, and (e) Lec domain. Each sAB concentration is shown in a different color trace. Within each plot, the

multiconcentration global fit line is shown in black. In order from highest to lowest, the concentrations of analyte used were 25, 12.5, 6.25, 3.125 nM. f SEC profiles and SDS-PAGE analyses show that the LK30 forms a monodisperse complex with the lectin domain of ADGRL3. g Binding activity of LK30 to the receptor was measured using HEK293T cells expressing full-length ADGRL3 (cyan curve) or full-length ADGRL1 (purple curve) by flow cytometry. $K_d$ values of LK30 binding were determined as 131 nM and 147 nM, for ADGRL3 and ADGRL1, respectively, by fitting the data to the concentration-response curve in GraphPad Prism. Cells transfected with empty vector were used as negative control (gray curve). Data are presented as mean ± SD of three repeats ($n = 3$) for a representative of three independent experiments. Source data are provided as a Source Data file.

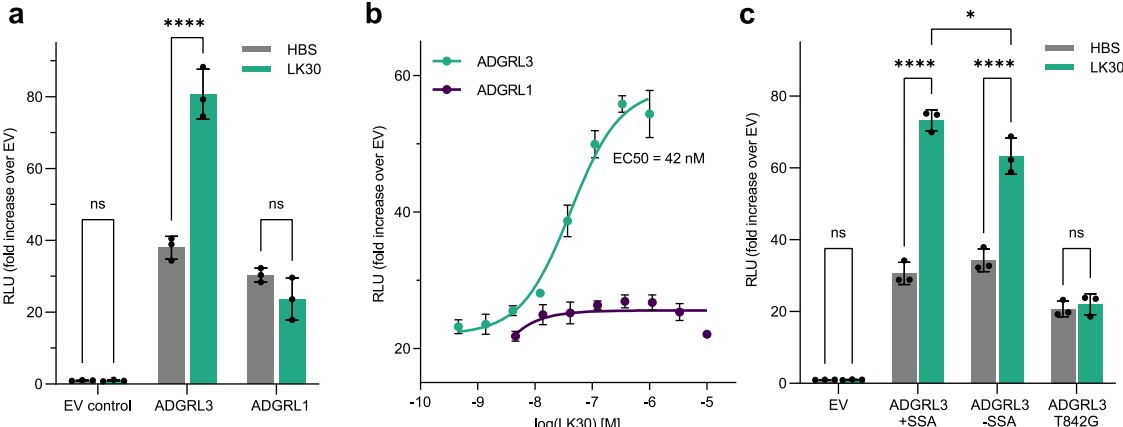

**Fig. 2 | LK30 is an ADGRL3-specific activator. a** SRE-luciferase assay for signaling of ADGRL3 and ADGRL1 in the absence or presence of 1 μM purified LK30 presented as fold increase over empty vector (EV). RLU, relative luminescence units. Data are presented as mean ± SD of three repeats (n = 3) for a representative of three independent experiments. $^{ns}p > 0.9999$; $^{****}p < 0.0001$; $^{ns}p = 0.6411$ vs. HBS buffer treatment; two-way ANOVA. **b** Titration of LK30 on SRE-luciferase activity of ADGRL3 (cyan) and ADGRL1 (purple). The EC50 value of LK30 on ADGRL3 was determined to be 42 nM. Data are presented as mean ± SD of three repeats (n = 3) for a representative of three independent experiments. **c** SRE-luciferase assay for the splice isoforms of ADGRL3 (+SSA and -SSA) and ADGRL3 autoproteolysis-null mutant (T842G, +SSA). Data are presented as mean ± SD of three repeats (n = 3) for a representative of three independent experiments, $^{ns}p > 0.9999$; $^{****}p < 0.0001$; $^{*}p = 0.0112$; $^{ns}p = 0.9992$ vs. HBS buffer treatment; two-way ANOVA. Source data for **a**–**c** are provided as a Source Data file.

activating it with its agonist, isoproterenol, we have shown that cAMP levels in cells expressing ADGRL3 or ADGRL1 were significantly lower compared to control (Supplementary Fig. 4). Therefore, we have tested the ADGRL3-activating LK30 in the cAMP assay. In contrast to the SRE-luciferase assay, LK30 showed no significant effect on ADGRL3- or ADGRL1-dependent cAMP levels suggesting that LK30 is a biased agonist for ADGRL3 (Supplementary Fig. 4).

### Mechanism of ADGRL3 activation by LK30 is autoproteolysis dependent

aGPCRs are cleaved in the conserved GAIN domain by an autoproteolytic mechanism[20]. Upon cleavage, the two fragments remain tightly associated[20]. Two complementary models for modulation of aGPCR activity by ligands have been proposed. In the Stachel-dependent model, ECR dissociation exposes the tethered agonist peptide (the Stachel peptide), allowing its direct interaction with 7TM domain and stimulating receptor activity[57–59]. Recent structural studies showed the direct interaction of the Stachel peptide with 7TM domain of multiple aGPCRs, including ADGRL3, and identified critical residues for receptor activation in the Stachel-dependent model[60–63]. GAIN domain autoproteolysis plays a crucial role here, allowing for the possible ECR dissociation, however self-cleavage has been shown not to be required in the case of ADGRF1 (GPR110)[62]. In the Stachel-independent model, ligand interaction with the ECR induces conformational changes, allowing for direct and transient interaction between the ECR and 7TM domain to activate the 7TM[21,22,24]. Contrary to the Stachel-dependent model, regulation occurring through this mechanism of aGPCR activation is independent of the receptor autoproteolysis.

We introduced a single-point mutation, T842G within the GAIN domain of ADGRL3, in order to establish whether LK30 activates ADGRL3 in an autoproteolysis-dependent or -independent manner (Supplementary Fig. 3a). We had previously shown that T to G mutation abolishes autoproteolysis within the GAIN domain without disrupting folding or cell-surface trafficking of the mutant receptor[20]. Using the SRE-luciferase assay, we first found that the basal activity of the autoproteolysis mutant is not affected when compared to the wild type receptor (Fig. 2c). LK30 treatment increased basal activity of ADGRL3 in HEK293T cells transfected with wild type ADGRL3; however, the effect of LK30 was nearly abolished when the cells were transfected with the ADGRL3 T842G mutant (Fig. 2c). This result

suggests that the agonistic effect of LK30 on the receptor signaling is dependent on the ADGRL3 autoproteolysis. Potentially, LK30-mediated activation of the ADGRL3 could lead to separation of the ECR from the 7TM upon LK30 binding. To test this hypothesis, we expressed N-terminally tagged ADGRL3 or ADGRL1 on the HEK293T cell surface and stimulated the receptors with the addition of LK30 to the media. We then analyzed both cell-lysate and corresponding media from each condition by western blot analysis and looked for changes in ECR levels after LK30 binding. No depletion of the ADGRL3 or ADGRL1 ECRs was observed in the cell lysate that was treated with LK30 compared to the cell lysates that was not (Supplementary Fig. 5a, c). Similarly, no ECR enrichment was observed in the media the LK30 treatment (Supplementary Fig. 5b, d). These results suggest that autoproteolysis, but not ECR separation from 7TM, is required for LK30-dependent activation of ADGRL3.

The ECR of ADGRL3 has a short alternatively-spliced region between Lec and Olf domains (a five amino acid splice insert: KVEQK) that was reported to decrease the affinity of ADGRLs to TENs when present[51]. Previous work had shown the importance of alternative splicing in regulating protein-protein interactions and functions[27,42]. In order to test the possible regulation of the LK30 effect on ADGRL3 signaling by alternative-splicing, we designed a ADGRL3 construct removing the five amino acids insert between Lec and Olf domains (ADGRL3 -SSA) and compared it to the construct that we have used throughout this study, ADGRL3 + SSA. LK30 treatment increased signaling of ADGRL3 -SSA with similar fold increase to the wild type ADGRL3 + SSA isoform, suggesting that the LK30 activation of ADGRL3 is not dependent on the splice isoform of the receptor (Fig. 2c).

### Crystal structure of ADGRL3/LK30 complex

To elucidate the molecular basis of the interaction between LK30 and ADGRL3 ECR, we determined the crystal structure of the Lec domain of ADGRL3 (ADGRL3$_{Lec}$) in complex with LK30 at 2.65 Å resolution (Fig. 3a, Supplementary Fig. 6a and Table 1). The crystal contacts in the structure are mediated predominantly by the heavy chain (HC) and light chain (LC) of the LK30, as well as by the Lec domains (Supplementary Fig. 6b). LK30 binds to Lec through CDRs in both HC (H1, H2, H3) and LC (L3), resulting in the total interface area of 608 Å² (HC – 558 Å²; LC – 50 Å²) in the protein complex. This interface is mediated by aromatic and hydrophobic residues of LK30 CDRs and involves extensive hydrogen bonding and van der Waals interactions with

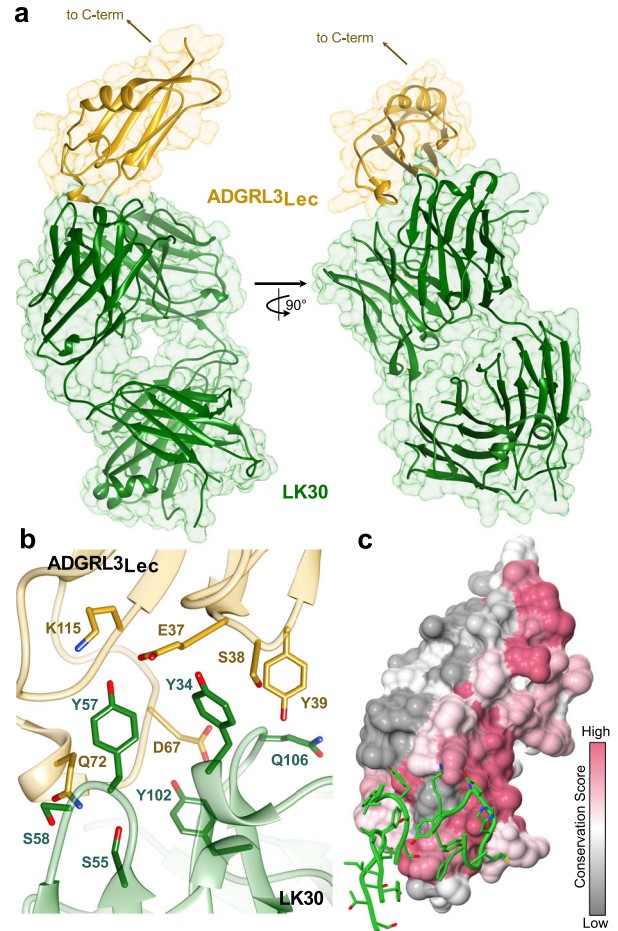

**Fig. 3 | Crystal structure of the LK30/ADGRL3 complex at 2.65 Å resolution.**
**a** The crystal structure of the ADGRL3 lectin domain in complex with LK30. ADGRL3$_{Lec}$ is colored yellow while the HC and LC of LK30 are colored green.
**b** Close-up view of the ADGRL3$_{Lec}$/LK30 interface. Residues at the binding interface are shown as sticks. ADGRL3$_{Lec}$ is colored yellow while the LK30 is colored green.
**c** Surface conservation analysis (gray, variable; red, conserved) of the ADGRL3 Lec domain. The CDRs of the LK30 HC interacting with the Lec domain are shown in green.

**Table 1 | Data collection and refinement statistics**

| ADGRL3$_{Lec}$/LK30 | |
|---|---|
| **Data collection statistics** | |
| Wavelength (Å) | 1.033 |
| Resolution range (Å) | 47.51–2.33 (2.40–2.33)[a] |
| Space group | P 2$_1$ 2$_1$ 2$_1$ |
| Unit cell dimensions (Å, °) | 81.80 97.85 163.06 90 90 90 |
| Total reflections | 108040 (8833) |
| Unique reflections | 56710 (4563) |
| Multiplicity | 1.9 (1.9) |
| Completeness (%) | 100 (99.9) |
| Mean I/sigma(I) | 9.3 (0.1) |
| Wilson B-factor (1/Å$^2$) | 80.04 |
| R-merge | 0.038 (7.31) |
| R-meas | 0.054 (10.34) |
| R-pim | 0.038 (7.31) |
| CC1/2 | 0.99 (0.079) |
| CC* | 1 (0.38) |
| **Refinement statistics** | |
| Resolution range (Å) | 47.51–2.65 (2.74–2.65) |
| Reflections used in refinement | 38700 (3768) |
| Reflections used for R-free | 3521 (352) |
| R-work | 0.24 (0.38) |
| R-free | 0.27 (0.40) |
| CC (work) | 0.94 (0.63) |
| CC (free) | 0.92 (0.52) |
| Number of non-hydrogen atoms | 8099 |
| Macromolecules | 8058 |
| Ligands | 34 |
| Solvent | 7 |
| Protein residues | 1059 |
| RMS (bonds) (Å) | 0.012 |
| RMS (angles) (°) | 1.56 |
| Ramachandran favored (%) | 95.39 |
| Ramachandran allowed (%) | 4.51 |
| Ramachandran outliers (%) | 0.10 |
| Rotamer outliers (%) | 1.86 |
| Clashscore | 7.74 |
| MolProbity Score | 1.95 |
| Average B-factor (Å$^2$) | 104.6 |
| Macromolecules | 104.6 |
| Ligands | 114.79 |
| Solvent | 72.43 |
| Number of TLS groups | 29 |

[a]Values in parentheses are for the highest-resolution shell.

residues in β1-β2 and β3-H1 loops at the tip of the Lec domain (Fig. 3b, Supplementary Fig. 6c). Notably, hydrogen bonds between E37$_{Lec}$ and Y34$_{LK30}$, Y39$_{Lec}$ and Q106$_{LK30}$, D67$_{Lec}$ and Y102$_{LK30}$, K115$_{Lec}$ and Y57$_{LK30}$ and Q72$_{Lec}$ and S55/S58$_{LK30}$ appear to stabilize the interaction and shape the total buried surface area.

## LK30 specifically breaks the interaction of ADGRL3 with TEN2, but not FLRT3

To visualize conserved and variable regions of the ADGRL3 Lec domain, we analyzed a heat map based on sequence conservation displayed (colored from most to least conserved) on the ADGRL3$_{Lec}$/LK30 complex structure (Fig. 3c, Supplementary Fig. 7a, b). We then mapped CDRs of the LK30 sAB (green sticks) on the ADGRL3$_{Lec}$ surface. Sequence conservation analysis revealed that LK30 binds to a highly conserved region on the N-terminal part of the Lec domain. As the ECR of ADGRLs has been previously shown to facilitate the interaction with its endogenous ligands−TENs and FLRTs (Fig. 4a), we hypothesized that LK30 binding might prevent those interactions. The structures of ADGRL bound to both TEN and FLRT have previously been reported[23,42,43,64]. We superimposed the ADGRL3$_{Lec}$/LK30 complex onto the structures of the ADGRL3/TEN2 and ADGRL3/FLRT3 complexes (Fig. 4b) showing the TEN binding site on the Lec domain of

ADGRL overlaps almost the entire sAB epitope. A more detailed analysis of the binding interfaces revealed that the LK30 binding site overlaps with the bottom part of the ADGRL/TEN complex interface (Fig. 4c). Notably, LK30 binding hinders one of the previously reported salt bridges−between D67 of ADGRL3 and K1712 of TEN2 from forming (Fig. 4c). Residue S38 of ADGRL3, which has been shown to interact with a conserved N-linked glycosylation on TEN2-N1681, is located in the buried hydrophobic pocket of sAB/Lec domain interface in the LK30 complex structure (Fig. 4c). Additionally, the LK30 binding blocks the interaction between the Lec domain and residues D1737 and H1738 of TEN2, that has been shown to be crucial for the ADGRL3/TEN2 interaction[42,43]. On the other hand, analysis of the FLRT binding

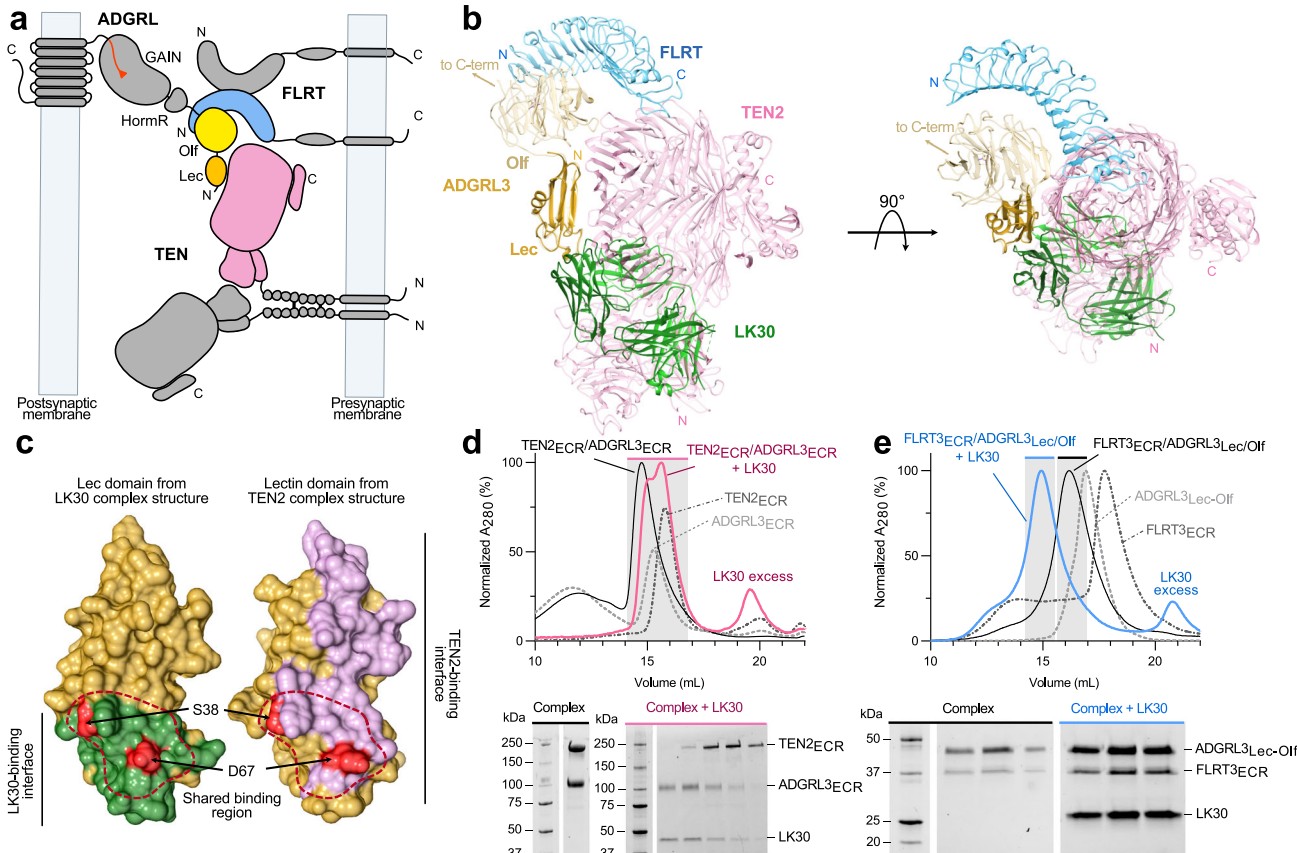

**Fig. 4 | LK30 blocks the interaction of ADGRL3 with TEN2 but not with FLRT3.**
**a** Schematic diagram of the interaction network between TEN, ADGRL, and FLRT at the synapse. TEN and FLRT are localized on the presynaptic cell membrane, while ADGRL is localized on the postsynaptic membrane. **b** Superimposition of the ADGRL3_Lec/LK30 complex with the ADGRL1/TEN2 complex structure (PDB: 6SKA) and trimeric complex of ADGRL3 and FLRT2 (PDB: 5FTU). The ADGRL domains of all structures are superimposed. ADGRL3_Lec and LK30 are colored yellow and green, respectively, TEN2 molecule is colored pink, FLRT is colored blue. The Lec and Olf domains bound to TEN2 and FLRT2 are colored tan. **c** Detailed analysis of ADGRL3 Lec domain regions interacting with either LK30 (left, in green) or TEN2 (right, in pink). Binding area shared by both ligands on Lec domain is indicated by a red dashed-line. Position of residues S38 and D67 shown as red patches. **d** SEC profiles showing the disruption of the ADGRL3/TEN2 complex (black curve) by the

LK30. Addition of the LK30 to the binary ADGRL3/TEN2 complex leads to TEN2 dissociation and formation of ADGRL3/LK30 complex, as observed on SEC profile (pink curve) and SDS-PAGE analysis. Fractions loaded on SDS-PAGE are highlighted by gray rectangle with colored line corresponding to the accompanying SDS-PAGE gel label. SEC profiles of individual proteins are shown as dashed curves. Results representative of three independent experiments. **e** SEC profiles showing the formation of the trimeric FLRT3/ADGRL3/LK30 complex (blue curve). Addition of the LK30 to the binary ADGRL3/FLRT3 (black curve) complex causes a shift in the retention volume on SEC profile. SEC profiles of individual proteins are shown as dashed curves. Fractions loaded on SDS-PAGE are highlighted by gray rectangle with colored line corresponding to the accompanying SDS-PAGE gel label. Results representative of two independent experiments.

site on the Olf domain of ADGRL3 suggests that LK30 interaction should not affect FLRT binding to ADGRL3. The remaining domains of ADGRL3 face away from the sAB, suggesting that LK30 would only interfere with TEN2 binding, but not with FLRT binding (Fig. 4b).

To test whether LK30 can prevent ADGRL/TEN complex formation, we performed SEC experiments with the ECRs of ADGRL3/TEN2. The addition of LK30 resulted in the dissociation of the ADGRL3/TEN2 complex, as observed on SEC chromatogram and SDS-PAGE analysis (Fig. 4d). The single peak of ADGRL3/TEN2 complex reforms into the complex of ADGRL3/LK30 (higher order species on SEC curve and first fractions of corresponding gel) and free TEN2 ECR (secondary peak on SEC curve and last three fractions of the gel). In order to measure the affinity of ADGRL3_Lec binding to TEN2 expressed on the cell surface, we transfected HEK293T cells with full-length TEN2 and used flow cytometry experiments to monitor binding of purified and biotinylated ADGRL3 Lectin domain. We determined the ADGRL3_Lec affinity to TEN2 as 129 nM (Supplementary Fig. 8a). We also performed a competition experiment in which we used a saturating concentration (5 μM) of Lec domain to bind to the TEN2 expressed on the cell surface. Then, we added increasing concentrations of the LK30 to pre-formed

ADGRL3_Lec/TEN2 complex to observe the dissociation of the Lec domain from TEN2 and measure the half maximal inhibitory concentration (IC50). LK30 disrupts Lec domain binding to TEN2 with IC50 of 193 nM (Supplementary Fig. 8b). We performed a similar set of experiments to test the ADGRL3 binding to FLRT3 (Fig. 4e). As expected from the structural analysis, we did not observe ADGRL3/FLRT3 complex dissociation. Instead, after addition of LK30 we observed a further shift in the retention volume, corresponding to the LK30/ADGRL3/FLRT3 trimeric complex formation as confirmed by SDS-PAGE analysis.

### sABs break specific intercellular contacts formed by ADGRL3-TEN2 and ADGRL3-FLRT3 interactions

Cell-adhesion proteins can interact with each other either in *cis*- or *trans*-. When expressed on the same cell surface, cell-adhesion proteins might be involved in *cis*-interactions. Alternatively, when cell-adhesion proteins are expressed on neighboring cells, they might be involved in *trans*-interactions that mediate cell-cell adhesion and intercellular contacts. Previous studies have shown that ADGRLs interact with TEN2 and with FLRT3 in a trans-cellular manner and

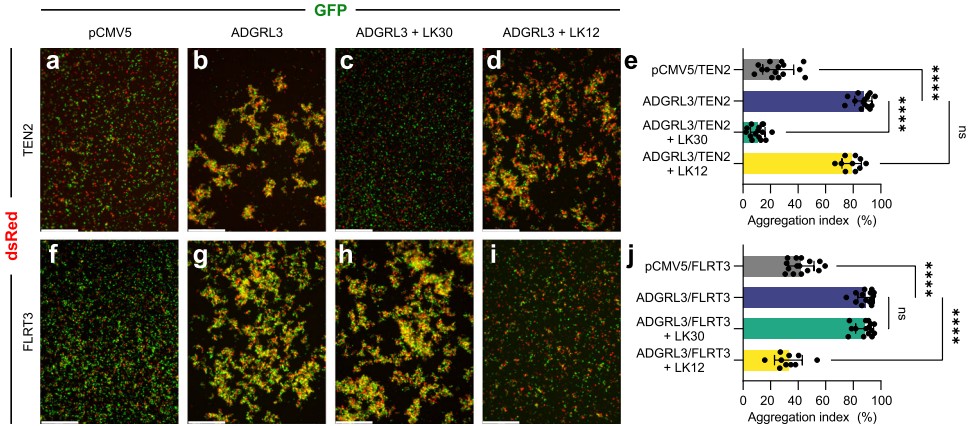

**Fig. 5 | sABs specifically inhibit cell-cell adhesion mediated by ADGRL3-ligand interactions. a-d** and **f-i** Cell-aggregation experiments show full-length ADGRL3 expressed on HEK293T cells interact with full-length TEN2 expressed on another population of HEK293T cells in a trans-cellular manner (**a, b**) Similarly, ADGRL3 interacts with full-length FLRT3 expressed on HEK293T cells in a trans-cellular manner as well (**f, g**). Addition of 5 μM sAB LK30 breaks the ADGRL3/TEN2 interaction and abolishes cell adhesion (**c**), but does not interfere with ADGRL3/FLRT3 interaction (**h**). In contrast, 5 μM of sAB LK12 breaks ADGRL3 interaction with FLRT3 (**i**), but not ADGRL3/TEN2-mediated cell adhesion (**d**). HEK293 cells were co-transfected with ADGRL3 or TEN2/FLRT3 and either GFP or dsRed as indicated. Scale bars: 500 μm. **e** and **j** Quantification of aggregation index are presented as mean ± SD from n = 15 images for LK30 experiments or n = 10 images for LK12 experiments, collected over three independent experiments, ****$p < 0.0001$; (e) $^{ns}p = 0.0539$; (j) $^{ns}p = 0.9902$; one-way ANOVA. Source data are provided as a Source Data file.

promote cell adhesion[23,51]. To examine the effect of LK30 on the interaction of ADGRL3 with its ligands, we performed cell aggregation assays with HEK293 cells in which each full-length protein is expressed on different cell populations and the cells are then mixed to monitor cell adhesion observed as cell-cell aggregation. As expected, when mixing cells expressing ADGRL3 with either TEN2- or FLRT3-expressing cells, we detected formation of significant cell aggregates when compared to control samples (Fig. 5a, b, e and f, g, j). Addition of LK30 to the mixtures significantly abolished ADGRL3/TEN2-mediated cell adhesion, validating our SEC results in the context of full-length receptors (Fig. 5c, e). LK30 binding had no effect on ADGRL3/FLRT3 interaction and cell aggregation suggesting that LK30 acts specifically on the ADGRL3-TEN2 interaction (Fig. 5h, j). Furthermore, we performed cell aggregation assays with the sAB LK12 that can recognize the Lec/Olf domains of ADGRL3 but not the Lec domain alone (Fig. 1b and Supplementary Fig. 9). As FLRT3 interacts with the Olf domain of ADGRL3, we speculated that LK12 might specifically affect ADGRL3-FLRT3 interaction. In contrast to LK30, LK12 abolished the ADGRL3 interaction with FLRT3, inhibiting ADGRL3/FLRT3-mediated cell adhesion, but did not affect ADGRL3 binding to TEN2 (Fig. 5d, i and e, j).

Taken together, these data show that we have developed highly specific sAB binders that can specifically block the ADGRL3 interaction with only one of its ligands, while preserving the interaction with the other.

## Discussion

aGPCRs are large chimeric molecules with transmembrane regions that are structurally homologous to the seven-pass GPCRs, and with ECRs that are homologous to cell-adhesion molecules (such as cadherins), receptor tyrosine kinases (such as EGF receptor) and others. Among the 33 human members of aGPCRs, 32 of them have ECRs and comprise numerous extracellular adhesion domains (ranging from one domain in ADGRG1/GPR56 to 40 domains in ADGRV1/GPR98) in addition to their conserved GAIN domain.

In this work, we employed the ADGRL subfamily of aGPCRs as a model system to demonstrate that the ECR of aGPCRs can be specifically targeted and funtionally modulated by antibodies. Utilizing phage display selection technology, we generated a number of ADGRL-specific synthetic antibodies. From this cohort, we focused on

characterizing LK30 and demonstrated it to be an activator of ADGRL3-dependent SRE signaling. We determined that LK30 binding to the N-terminal Lec domain of the ADGRL3, distal from the 7TM region, increases the basal signaling of the receptor. To further investigate LK30-dependent stimulation, we tested the effect of autoproteolysis within the GAIN domain on the receptor modulation and found that the specific cleavage at the GPS site, but not ECR dissociation is required for the ADGRL3 activation by LK30. These results, along with our previously published work on Stachel-independent modulation of ADGRG1 signaling[22], present two different mechanisms of ECR-targeted aGPCR activation: autoproteolysis-dependent for ADGRL3 and autoproteolysis-independent for ADGRG1.

When targeting receptors, it is important to consider different isoforms (paralogs which have been generated as a result of gene duplication) and splice variants of the gene. Each isoform may have evolved to facilitate different functions[65–67] and specific targeting of a particular form can lead to different functional readouts. Similarly, the importance of alternative splicing in regulating protein-protein interactions and differentiating protein function has been widely studied[1,27,42]. In this regard, the LK30 sAB shows that antibodies can act in an isoform-specific manner. The SRE-signaling assays revealed that LK30 acts as an agonist for ADGRL3. However, it does not activate ADGRL1, although it can bind to both isoforms with similar affinity (Fig. 6a). In addition, LK30 acts as a biased agonist for ADGRL3 as it is active only in the SRE assay but not the cAMP assay. This further provides an example where antibody-mediated modulation of receptor signaling can be unique for different isoforms and different signaling pathways. Though the mechanism of this isoform specificity is not known, we speculate that the explicit change of the ECR conformation necessary to allosterically activate the receptor cannot be induced by LK30 for ADGRL1, due to differences in the ECR sequence (Supplementary Fig. 7c). In contrast to isoform-specificity, LK30 activation does not appear to be dependent on ADGRL3 splice site insertion. As for the case of aGPCRs, the specific functions of aGPCR isoforms are still under investigation. However, there is growing evidence that the different ADGRL isoforms are critical for synapse formation at different sublocations within the same neuron[40,48]. LK30 or similarly specific antibodies have the potential to specifically modulate certain types of synapse formation in a single neuron.

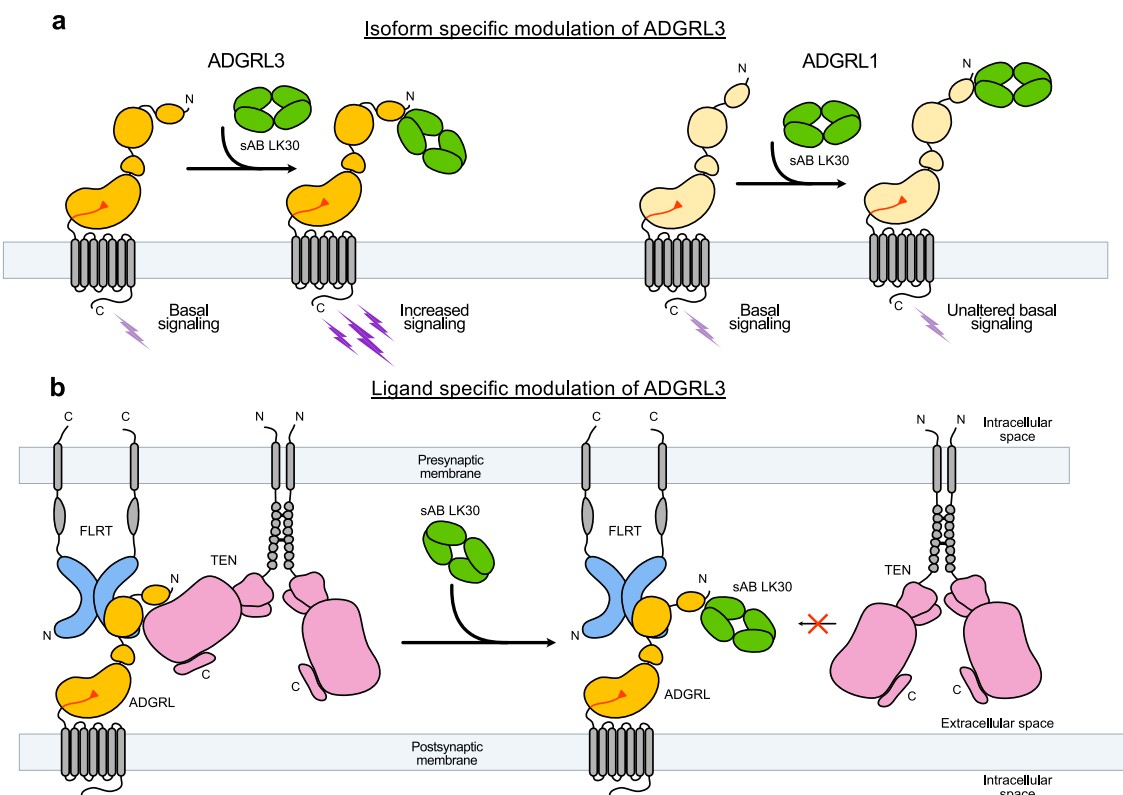

**Fig. 6 | LK30 modulates ADGRL3 in an isoform- and ligand-dependent manner. a** Binding of LK30 to ADGRLs modulates the receptor activity in an isoform-specific manner. ADGRL3 (dark yellow) basal signaling increases upon binding of LK30 to the Lec domain of the receptor. The interaction of LK30 with ADGRL1 (light yellow) does not change the signaling activity in the SRE assay. **b** LK30 breaks the interaction of ADGRL3 with TEN2, whereas it has no effect on the interaction of ADGRL3 with FLRT3. Synthetic antibodies can be used to specifically target and block interactions of the receptor with its endogenous ligand.

The crystal structure of the LK30 in complex with Lec domain of ADGRL3 revealed the LK30 binding site. Based on comparisons with other available ADGRL complex structures, the LK30 epitope partially overlaps with the TEN binding site. We showed that the LK30 can block the interaction of ADGRL with TEN, while maintaining its interaction with FLRT (Fig. 6b). Such a sAB can be utilized to abolish TEN-dependent functions of ADGRLs such as excitatory synapse formation and cAMP signaling as previously reported[5,26]. Additionally, utilizing another sAB – LK12 that inhibits ADGRL3/FLRT3 complex formation, we demonstrate that synthetic antibodies can be utilized to specifically target and block interactions of the receptor with only one of its endogenous ligands.

Although most aGPCRs are still orphan receptors with no known ligands, it is reasonable to suppose that other aGPCRs also have multiple ligands that bind to the ECR and modulate receptor function. For instance, ADGRG1 has three known binding partners, each contributing to one of its three different functions: the interaction of ADGRG1 with collagen 3 mediates brain cortex development, while tissue transglutaminase 2 mediates central nervous system myelination and phosphatidylserine mediates microglia activation[68,69]. Drugging aGPCRs will thus, require a sophisticated approach rather than simply turning on or off their downstream signaling by targeting the 7TM. Indeed, it has been reported that the small molecule α-DOG which binds to the 7TM of an ADGRG1 and acts as an agonist, also affects another aGPCR ADGRG5/GPR114[31]. Because of the high variation and diversity of aGPCR ECRs, the development of ECR-targeted ligands is more likely to result in highly specific reagents for aGPCRs. An additional advantage of targeting the ECR is gaining the ability to disrupt the interactions between aGPCRs and their ligands in a specific manner. Thus, precise targeting of aGPCRs by synthetic molecules in a ligand- and isoform-specific manner may be used as a foundation for drug design to treat aGPCR-mediated diseases.

## Methods
### Protein expression and purification
The DNA constructs for ECRs of ADGRL3[23,42], TEN2[26,42] and FLRT3[23] expression and purification were published previously. All proteins were expressed using the baculovirus method. *Spodoptera frugiperda* Sf9 insect cells (Thermo Fisher, 12659017) were co-transfected with the constructed plasmids and linearized baculovirus DNA (Expression Systems, 91-002) using Cellfectin II (Thermo Fisher, 10362100). Baculovirus was amplified in Sf9 cells in SF-900 III medium supplemented with 10% FBS (Sigma–Aldrich, F0926).

ECR constructs of ADGRL3, TEN2 and FLRT3 were expressed in High Five cells (Thermo Fisher, B85502). Cell cultures grown in Insect-Xpress medium (Lonza, 12-730Q) were infected with high-titer baculovirus at a density of $2.0 \times 10^6$ cells ml$^{-1}$ and incubated for 72 h at 27 °C. The cells were pelleted by centrifugation and the conditioned medium containing the secreted glycosylated proteins were collected. Final concentrations of 50 mM Tris pH 8, 5 mM CaCl$_2$, and 1 mM NiCl$_2$ were added to the media, the mixture was then stirred for 30 min and centrifuged at $8000\,g$ for 30 min to remove the precipitate. The supernatant was incubated with Ni-NTA resin (Qiagen, 30250) for 3 h. The resin was collected with a glass Buchner funnel and washed with 10 mM HEPES pH 7.2, 150 mM NaCl and 20 mM imidazole. Avi-tagged proteins were then biotinylated on-column, incubating the resin-bound proteins with 50 mM bicine pH 8.3, 10 mM MgOAc, 100 mM NaCl, 10 mM ATP, 0.5 mM biotin and 5 mM BirA at 27 °C with gentle mixing. The protein sample was later eluted with 10 mM HEPES pH 7.2, 150 mM NaCl and 200 mM imidazole. The elution fractions were concentrated and loaded onto a Superdex S200 10/300 GL column (GE

Healthcare) equilibrated with 10 mM HEPES pH 7.2 and 150 mM NaCl. Purified fractions of the protein were collected for further experiments.

## Phage Display Selection

Selection for ECR fragments of ADGRL3 was performed according to previously published protocols[55,70]. 200 nM of target was immobilized on streptavidin magnetic beads for the first round of selection. Next, the beads were washed three times to remove unbound protein and 5 mM D-biotin was added to block unoccupied streptavidin on the beads to prevent nonspecific binding of the phage. Afterwards, the beads were incubated for 30 min at RT with the phage library E[71], containing $10^{12}$–$10^{13}$ virions ml$^{-1}$ with gentle shaking. This was followed by washing of beads containing bound phages, which were later used to infect log phase *E. coli* XL1-Blue cells. Infected cells were grown overnight in 2YT media with 50 µg/mL ampicillin and $10^9$ p.f.u. ml$^{-1}$ of M13 KO7 helper phage in order to amplify phages. Three additional rounds of selection were performed with decreasing target concentration in each round (100 nM, 50 nM, 10 nM) using the amplified pool of virions of the prior round used as the input. Rounds 2 to 4 were performed using semi-automated platform using the Kingfisher instrument. In those rounds phages were eluted using 0.1 M glycine pH 2.7. This technique often risks the enrichment of nonspecific and streptavidin binders. In order to eliminate them, the precipitated phage pool from rounds 2 to 4 were negatively selected against 100 mL of SA beads. The "precleared" phage pool was then used as an input for the selection.

## Enzyme-linked immunoabsorbent assays (ELISA)

ELISA experiments were carried out using a 96-well flat-bottom plate coated with 50 µL of 2 mg ml$^{-1}$ neutravidin in $Na_2CO_3$ pH 9.6 and subsequently blocked with 0.5% Bovine Serum Albumin in 1 × PBS. Binding screens of all of the selected sABs in phage format was performed using a single point phage ELISA. A total of 400 µL of 2YT media with 100 µg ml$^{-1}$ ampicillin and M13 KO7 helper phage were inoculated with single *E. coli* XL1-Blue colonies harboring phagemids, and cultures were grown at 37 °C for 18 h in a 96-deep-well block plate. The cells were pelleted by centrifugation and sAB phage-containing supernatants were diluted 20× in ELISA buffer. Diluted phages were then applied to ELISA plates, preincubated for 15 min with 50 nM of biotinylated target proteins at RT. Plates with added phages were incubated for 15 min at RT and washed 3-times with 1×PBST. The washing step was followed by 30 min incubation with HRP-conjugated Mouse Anti-M13 monoclonal antibody (GE Healthcare, 27942101) diluted in PBST in 1:5000 ratio. Excess antibody was washed away with 1 × PBST and plates were developed using TMB substrate, quenched with 1.0 M HCl and the signal was determined by absorbance measurement ($A_{450}$).

Protein based single-point ELISA was performed to confirm binding of generated unique sABs to their cognate antigens. Immobilized on ELISA plate target (50 nM) was incubated with 200 nM of the purified sABs for 15 min. The plates were then washed and incubated with a secondary HRP-conjugated anti-human F(ab')$_2$ monoclonal antibody (Jackson ImmunoResearch, 109-035-006; 1:5000 dilution in PBST). The plates were washed, developed with TMB substrate and quenched using 1.0 M HCl, and absorbance ($A_{450}$) signal was measured.

## Cloning, overexpression and purification of sABs

Phage ELISA results were used to select sAB clones that were sequenced at DNA Sequencing Facility at The University of Chicago. In-fusion cloning[72] was used to reformat unique sABs clones into pRH2.2, an IPTG inducible vector for bacterial expression.

*E. coli* BL21 (Gold) cells were transformed with sequence-verified sAB plasmids. Cultures were grown in 2YT media supplemented with 100 µg mL$^{-1}$ of ampicillin at 37 °C until they reached $OD_{600}$ = 0.8, when they were induced with 1 mM IPTG. The culture was continued for 4.5 h at 37 °C and cells were harvested by centrifugation. Cell pellets were resuspended in 20 mM HEPES pH = 7.5, 200 mM NaCl, 1 mM PMSF, 1 µg ml$^{-1}$ DNase I, and lysed by ultrasonication. The cell lysate was incubated at 60 °C for 30 min. Heat-treated lysate was centrifuged at 50,000 x g to remove cellular debris, filtered through a 0.22 µm filter and loaded onto a HiTrap protein L (GE Healtchcare) column pre-equilibrated with 20 mM HEPES pH 7.5 and 500 mM NaCl. The column was washed with 20 mM HEPES pH 7.5 and 500 mM NaCl and sABs were eluted with 0.1 M acetic acid. Protein-containing fractions were loaded directly onto an ion-exchange Resource S column pre-equilibrated with 50 mM NaOAc pH 5.0 and washed with the equilibration buffer. sABs elution was performed with a linear gradient 0–50% of 50 mM NaOAc pH 5.0 with 2 M NaCl. Purified sABs were dialyzed overnight against 20 mM HEPES pH 7.5 with 150 mM NaCl. The quality of purified sABs was analyzed by SDS-PAGE.

## Binding Kinetics by SPR

SPR experiments were performed at RT using MASS-1 instrument (Bruker) with a His-capture sensor chip (XanTec). Targets were immobilized onto a nitrilotriacetic acid (NTA) sensor chip via His-tag. For kinetic experiments, 2-fold serial dilutions of the sAB were injected following ligand immobilization on the sensor chip. For kinetic assay four to six dilutions of the sAB were tested. Following each injection, the chip surface was regenerated using 350 mM EDTA and 50 mM NaOH, and ligands were subsequently immobilized for the following injection. For each ligand, all experiments were performed on single channels for consistency. The double-reference subtracted curves were then fit with one-to-one Langmuir binding models in Scrubber to determine kinetic binding parameters.

## Multipoint protein ELISA

For multipoint ELISA a fixed concentration of the immobilized target (50 nM) on ELISA plate was incubated with 3-fold serial dilutions of the purified sAB starting from 1 µM for 15 min. The plates were washed, and the bound antigen-sAB complexes were incubated with HRP-conjugated anti-human F(ab')2 monoclonal antibody (Jackson ImmunoResearch, 109-035-006; 1:5000 dilution in PBST) for 30 min. Subsequently the plates were washed again, developed with TMB substrate and quenched with 1.0 M HCl, and absorbance ($A_{450}$) was determined. Affinity was determined by fitting in a non-linear sigmoidal function with variable slope in GraphPad Prism 9.3.1 and $EC_{50}$ value was calculated.

## Flow cytometry

HEK293T cells (ATCC, CRL-3216) were seeded in 6-well plates with DMEM (Gibco, 11965092) supplemented with 10% (v/v) FBS (Sigma-Aldrich, F0926). After 24 h, cells reached 50-60% confluence and were transfected with 1 µg of ADGRL3, ADGRL1[44], or TEN2[42] or EV using 6 µl of transfection reagent LipoD293T. After 48 h, cells were detached with citric saline solution and washed with 0.1 % BSA in PBS. For cell surface expression measurements, cell pellets were incubated for 2 h with 1:1000 dilution of mouse anti-FLAG antibody (Sigma-Aldrich, F3165), washed three times with 1xTBST and incubated with 1:500 dilution of secondary AlexaFluor488-conjugated Donkey Anti-Mouse antibody (Invitrogen, A21202). For LK30 binding experiments, the pellets were incubated with increasing concentrations of LK30 for 30 mins at 4 °C, washed twice with 0.1 % BSA in PBS and incubated with fluorescent-tagged secondary antibody (1:1000 dilution; Alexa Fluor 647 AffiniPure Goat Anti-Human IgG, F(ab')$_2$ fragment specific; Jackson ImmunoResearch #109-605-006) for 30 min at 4 °C, washed three times and fixed in 1% paraformaldehyde. To test ADGRL3/TEN2 interaction, biotinylated Lec domain of ADGRL3 was incubated with cells expressing FL TEN2. ADGRL3$_{Lec}$ binding to the TEN2 on cell surface was monitored by addition of fluorescently labeled neutravidin

(NeutrAvidin Protein, DyLight™ 650, Invitrogen #84607) and binding affinity was estimated by fitting the concentration-response curve in GraphPad Prism 9.3.1. Then, using saturating concentration (5 μM) of Lec domain of ADGRL3 bound to the TEN2 on cell surface, the LK30 sAB was added in increasing concentrations to estimate the IC50. Flow cytometry data were collected on BD Accuri C6 flow cytometer and analyzed in FlowJo.

## Serum response element luciferase assays

Human ADGRL3 T842G mutant was generated using the QuikChange method (Agilent) with primers: F: 5′-ACATGCAGCTGTAATCACC TGGGCAACTTTGCTGTCCT-3′ and R: 5′-AGGACAGCAAAGTTGCCC AGGTGATTACAGCTGCATGT-3′. The ADGRL3 -SSA construct was generated using primers: F: 5′-TACGAGTGCGTGCCATATGTGTTCCT GTGCCCCGGCCT-3′ and R: F: 5′- AGGCCGGGGCACAGGAACACAT ATGGCACGCACTCGTA-3′. HEK293T cells (ATCC, CRL-3216) were seeded on a 96-well flat-bottom plate precoated with 0.5% gelatin and grown until 50–60% confluent in DMEM supplemented with 10% (v/v) FBS. Cells were then co-transfected with ADGRL3/ADGRL1 constructs (1 ng well$^{-1}$ for ADGRL3; 5 ng well$^{-1}$ for ADGRL1)[44], Dual-Glo luciferase reporter plasmid (20 ng well$^{-1}$)[21], using 0.3 μL LipoD293T (SL100668; SignaGen Laboratories). DNA levels were balanced among transfections by addition of the empty pCMV5 vector to 100 ng total DNA. Eighteen hours after transfection media was aspirated and replaced with DMEM without FBS. When sABs were tested, 1 μM of sAB was added 5 h after start of the serum-starvation. After 10 h of serum starvation, the media was removed and cells were lysed using Dual-Glo Luciferase Assay System from Promega and firefly and renilla luciferase signals were measured using a Synergy HTX (BioTek) luminescence plate reader. The firefly:renilla ratio for each well was calculated and normalized to empty vector. Data were then analyzed using GraphPad Prism 9.3.1 software and presented as mean ± SD of three repeats ($n = 3$) for a representative of three independent experiments.

## cAMP assay

HEK293 cells (ATCC CRL-1573) were seeded in 6-well plates with DMEM supplemented with 10% (v/v) FBS. After 24 h, cells reached 50-60% confluence and were transfected with 350 ng of ADGRL3/ADGRL1, 350 ng of 22 F Glosensor reporter plasmid (E2301; Promega) and 9 ng of β2-adrenergic receptor using 2.8 uL of transfection reagent Fugene 6 (Promega, PRE2693). After 24 h, cells were detached and seeded at $5 \times 10^4$ per well in white flat bottom 96-well plate. After another 24 h, the media was replaced with 100 μl Opti-MEM I Reduced-Serum Medium (31985070, Life Technologies) and incubated for 30 min. Then, 1 μL of Glosensor substrate and 11 μL of FBS were added to each well. Basal cAMP signal was measured after 20 min of equilibration time. Next, cells were treated with 2 μM of sAB for 5 min and then activated with 50 nM of isoproterenol. Measurements were done using Synergy HTX BioTek plate reader at 25 C. Data were then analyzed using GraphPad Prism 9.3.1 software and presented as mean ± SD of three repeats ($n = 3$) for a representative of three independent experiments.

## Western blot

HEK293T cells (ATCC, CRL-3216) were seeded in 6-well plates in DMEM supplemented with 10% (v/v) FBS and grown until 80–90% confluent. Then, cells were transfected with 2 μg of ADGRL3/ADGRL1 constructs. After 48 h post transfection, the LK30 or HBS was added to the media and incubated for 5 h. Then, media was collected and cells were washed with 1 × PBS supplemented with 0.01% BSA, harvested and solubilized with 500 μL of 20 mM HEPES pH = 7.4, 150 mM NaCl, 2 mM MgCl$_2$, 0.1 mM EDTA, 2 mM CaCl$_2$ and 1% (v/v) Triton X-100 for 30 min at 4 °C. Samples were then spun down at 20,000 × $g$, and supernatants were collected. Solubilized cell supernatants and media samples were run on a 4−20% SDS-PAGE gel (Biorad, #4561096), followed by Western

blotting. Membranes were blocked for 1 h at room temperature with 4% BSA in TBST buffer and washed three times with TBST buffer. Then, the membranes were incubated overnight at 4 °C with the antibodies (1:1000 dilution; THE™ DYKDDDDK Tag Antibody [iFluor 488], mAb, Mouse; Genescript #A01809 or Alexa Fluor 647 AffiniPure Goat Anti-Human IgG, F(ab')$_2$ fragment specific; Jackson ImmunoResearch #109-605-006). The next morning, membranes were washed with TBST buffer and visualized using the Bio-Rad Gel Imager.

## Formation of ADGRL3$_{Lec}$/LK30 complex

ADGRL3/sAB complex was formed by mixing 1.5-fold molar excess of the Lec domain with the LK30 sAB and 30 min incubation on ice. Next, the complex was subjected to size-exclusion chromatography on a Superdex 200 10/300 column pre-equilibrated with 30 mM HEPES pH 7.5 with 150 mM NaCl. Formation of the ADGRL3$_{Lec}$/LK30 complex was determined by retention volume analysis of the complex with respect to that of target alone and co-elution of the individual components on SDS-PAGE.

## X-ray crystallography

Purified as described above ADGRL3$_{Lec}$/LK30 complex was crystallized using hanging-drop vapor diffusion at 5 mg mL$^{-1}$ in 100 mM sodium acetate (pH = 4.5), 150 mM ammonium sulfate and 20% (w/v) PEG 4000. Crystals were frozen in mother liquor with the addition of 20% glycerol. Data were collected to 2.33 Å at the Advanced Photon Source at Argonne National Laboratory (beamline 23-ID-B). The datasets were auto processed using the GM/CA beamline GMCAproc protocol, which employs XDS[73] and pointless[74] to index, integrate, scale, and merge the data. Initial phases were determined by molecular replacement using PHASER[75] in the CCP4 7.1.016 suite[76], utilizing the available human Lec domain structure (PDB ID: 6VHH)[42] and sAB structure (PDB ID: 4XWO)[77] as search models. The structure of the complex was obtained in space group P2$_1$2$_1$2$_1$ with two ADGRL3$_{Lec}$/LK30 complexes in the asymmetric unit. Initial rounds of refinement and model building were performed with REFMAC5 (CCP4 7.1.016) using NCS restraints. Next, phenix.refine (PHENIX 1.19.2-4158)[78] was used with d$_{min}$ = 2.65 Å without NCS restraints but with reference model restraints, using high resolution structures of the Lec domain (PDB 5AFB)[79] and sAB (PDB 5UCB). Final rounds of model building and refinement were performed in COOT 0.9.6 and phenix.refine without reference model restraints or NCS restraints. Final refinement parameters were rigid body refinement with individual B-factors, TLS refinement, and optimization of stereochemistry and ADP weighting. In the structure we were able to assign three types of ligands: PEG, glycerol and sulfate molecules, all present in the crystallization solution.

## Conservation analysis

For the conservation analysis, the ConSurf server was utilized using the software default parameters[80,81]. 150 sequences that sample the list of homologs to reference (human ADGRL3$_{Lec}$) was searched using HMMER search algorithm in UNIREF-90 database. Sequences with minimum of 35% and maximum 95% identity were used. After aligning with MAFFT-L-INS-i method, the conservation was calculated using Bayesian method and mapped to the surface of ADGRL3$_{Lec}$ in UCSF Chimera.

## LK30 binding to ADGRL/TEN and ADGRL/FLRT complexes

For ADGRL3/TEN2/LK30 complex test, ECRs of ADGRL3 and TEN2 were co-expressed in High Five insect cells, purified by Ni-NTA affinity chromatography and subjected to SEC. Purified fractions were pulled and 2-fold molar excess of LK30 was added, followed by 30 min incubation on ice. Next, the mixture was subjected to SEC on a Superose 6 10/300 column pre-equilibrated with 30 mM HEPES pH 7.5 with 150 mM NaCl.

For AGRL3/FLRT3/LK30 tests, ECRs of ADGRL3 and FLRT3 were expressed in High Five insect cells. After purification of individual

proteins, the complex was formed by mixing proteins in 1:1 molar ratio at room temperature for 30 min, followed by SEC purification. Then, 2-fold molar excess of LK30 was added, the mixture was incubated for 30 min on ice and injected on Superdex 200 10/300 column pre-equilibrated with 30 mM HEPES pH 7.5 with 150 mM NaCl.

### Cell Aggregation Assay

HEK293T cells (ATCC, CRL-3216) were seeded in 6-well plate containing 2.5 mL of DMEM media supplemented with 10% FBS and incubated overnight at 37 °C. When cells reached 80% confluency, they were co-transfected with 2 µg of either pCMV5 + GFP, pCMV ADGRL3 + GFP, TEN2 + dsRed, or FLRT3 + dsRed using 4 µL LipoD293T (SL100668; SignaGen Laboratories). Two days after transfection, the media was aspirated, cells were washed with 1xPBS and detached with 1xPBS containing 1 mM EGTA and supplemented with 15 µL of 50 mg ml$^{-1}$ DNAse I (Sigma, D5025). Cells were resuspended by pipetting to a create single-cell suspensions, transferred to an Eppendorf tube and additional 15 µL of DNAse solution was added to each sample. Seventy µL of cells expressing indicated constructs were mixed in 1:1 ratio in a one well of a non-coated 24-well plate containing 340 µL of Incubation Solution (DMEM supplemented with 10% FBS, 10 mM CaCl$_2$ and 10 mM MgCl$_2$). Forty µL of either sAB LK30 (to final concentration of 5 µM) or 1×HBS was added to the mixture, and cells were then placed on a shaker at 120 rpm at 27 °C for 10 min and imaged using a Leica Fluorescent Microscope with a 5× objective. Aggregation index at time = 10 min was calculated using ImageJ 1.52e. A value for particle area of 1–2 cells was set as a threshold based on negative control values. The aggregation index was calculated by dividing the area of particles exceeding this threshold by the total area occupied by all particles in the individual fields.

### Reporting summary

Further information on research design is available in the Nature Portfolio Reporting Summary linked to this article.

## Data availability

The coordinates for the crystal structure of ADGRL3/LK30 generated in this study have been deposited in the Protein Data Bank [http://www.rcsb.org] under accession code PDB 8DJG. All the other relevant structures referenced in this work are available under the accession codes 6VHH, 4XWO, 5AFB, 5UCB, 6SKA and 5FTU. The authors declare that all data supporting the findings of this study are available within the article and the source data underlying Figs. 1b, g, 2a-c, 5e, j and Supplementary Figs. 4, 5 and 8 are provided as a Source Data file. The gating strategy for flow cytometry experiments can be found in Supplementary Fig. 10. Source data are provided with this paper.

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

## Acknowledgements

We thank Engin Özkan for the use of flow cytometer and all members of the Araç lab for helpful discussions. We also thank the staff at the Advanced Photon Source (APS) at Argonne National Labs (ANL). GM/CA@APS has been funded by the National Cancer Institute (ACB-12002) and the National Institute of General Medical Sciences (AGM-12006, P30GM138396). This research used resources of the Advanced Photon Source, a U.S. Department of Energy (DOE) Office of Science User Facility operated for the DOE Office of Science by Argonne National Laboratory under Contract No. DE-AC02-06CH11357. The Eiger 16 M detector at GM/CA-XSD was funded by NIH grant S10 OD012289. This work was supported by grants R35 GM148412 (to D.A.), R01 GM134035-01 (to D.A.), F32 GM142266 (to S.J.B.) and R01 GM117372 (to A.A.K.).

## Author contributions

S.P.K. and D.A. designed all experiments and interpreted results. S.P.K. expressed and purified all proteins (with assistance from B.A. and K.L.). P.D. carried out phage display selection and sABs characterization. S.K.E. and S.P.K. assisted with SPR experiments. S.P.K. and J.M.A. performed flow cytometry and cell-based signaling assays. S.J.B. and S.P.K. performed crystallography experiments (with assistance from B.A.) and structure determination. J.L. participated in cell-cell aggregation experiments. S.P.K. and D.A. wrote the manuscript with assistance from A.A.K., S.J.B. and P.D.. D.A. and A.A.K. supervised the project.

## Competing interests

The authors declare no competing interests.
