## [Peer Review File · Nature Communications]

REVIEWER COMMENTS

Reviewer #1 (Remarks to the Author):

Comments to Authors:

This paper describes the development of a novel antibody that targets the lectin domain of ADGRL3. The figures are beautiful, and the experiments are well described.

There were some specific questions, however, that might benefit from some further clarification.

1) The authors write that antibodies targeting adhesion GPCRs have been developed before, i.e., Ref 22, Ref 28, and Ref 29. It is not clear how LK30 is different and a key advancement to the field. It would be helpful to spell this out for readers.

- * is the binding site different/similar/same?
- * is the affinity different/better/worse?
- * are the other antibodies agonists or antagonists?
- * is the selectivity different/similar/same towards ADGRs?

2) The authors write that LK30 is unique because it is selective for ADGRL3 and inactive against ADGRL1. This finding is a bit puzzling.

LK30 binds ADGRL3 quite tightly, $KD \sim 4$ nM using SPR (i.e., biochemically). This reviewer could not find where it was stated how well LK30 binds ADGRL1 by SPR. LK30 binds ADGRL1 with $KD \sim 131$ nM and ADGLR1 ~ 147 nM, i.e. roughly similarly, by flow cytometry.

Yet, LK30 is active against ADGRL3 (agonist) but not against ADGRL1 (inactive) in a cell-based assay (SRE-luciferase). LK30 is inactive against ADGRL3 using a cAMP based cell-assay, but this reviewer could not find whether LK30 is active against ADGRL1 in the cAMP based-assay.

To make a solid case that i) LK30 is selective (active against ADGRL3 and not ADGLR1), and ii) works allosterically (see Discussion), it would greatly strengthen the paper to fill in the missing information:

- * show by SPR the binding affinity of LK30 for ADGRL1, as well as ADGRL3
- * make a sequence alignment comparing ADGRL3 and ADGRL1 in the region where ADGRL3 binds to LK30 (its CDR loops, Fig. 3b and Fig. 3c). Fig. 3c lacks a description of what sequences were compared so it is hard to assess. Also the conservation score in Fig. 3c is hard to assess. Which residues are identical? Which residues are semi-conserved and what residues were grouped together to form 'semi-conserved' designations?
- * explain why LK30 is active ADGRL3 in a SRE-luciferase assay (but ADGRL1 is not); while it is not active against ADGRL3 in the cAMP based assay, and state whether LK30 works against ADGRL1 in the cAMP based assay.

3) line 129: protein L affinity,
Please define

4) line 134: while one sAB recognized the Olf domain
Which sAB? Experiment shows that Olf domain or both Olf-Lec domains are recognized, correct? If so, please revise

5) line 159: SRE-luciferase assay

It is very helpful for the reader to quickly explain how this assay works to assess ADGRL1/3 activity.

6) line 165-168: despite ability of ADGRL1 to bind LK30 (Fig. 2A, B)

On what basis does ADGRL1 bind to LK30? See also point 2).

7) line 169-170: cAMP-based assay

It is very helpful for the reader to quickly explain how this assay works to assess ADGRL1/3 activity.

8) line 172-173: LK30 has no effect on ADGR3 in cAMP assay

Please explain why this might be.

Please indicate if LK30 is active against ADGRL1 in the cAMP assay.

9) line 176: recent studies and models for autoproteolytic mechanism

Perhaps, the authors would like to revise lines 176-182 to include recent studies, e.g., the recent four studies summarized in Boucard (2022) Nature Vol 604, p. 628

10)line 218-219: conserved region on N-term part of Lec

Please indicate in the Material and Methods and in the legend for Fig. 3c, which sequences were used for the multi-sequence alignment.

Please indicate if sequences for ADGRL3 and ADGRL1 both were compared or only for ADGRL3.

Please add a version of the figure where the sequence conservation between ADGRL3 and ADGRL1 in the Lec domain is shown.

Please show which residues are 100% identical and which are semi-conserved. Please also indicate how residues are grouped for the semi-conserved analysis.

Please include the multi-sequence alignment as a supplemental figure.

11)line 227-228: D67 and S38

Please indicate in Fig. 4C (with little arrows) where D67 and S38 are.

Please indicate in Fig. 3B (label) where S38 is.

12)line 229: buried in hydrophobic pocket in the LK30

Please indicate ".....buried in hydrophobic pocket of ??whom??. in the LK30....."

13) line 242-244

It is helpful to explain to the reader why disruption of the ADGRL3-teneurin complex results in increased activity in the SRE-luciferase assay.

14)line 259: "of ADGRL3/FLRT3 interaction"

Should this read "on ADGRL3/FLRT3 interaction...."?

15)line 261: ADGRL3 Olf domain binder

Shouldn't this be Olf domain or Olf-lec tandem? Sometimes antibodies can recognize epitopes coming from different domains.

16)There is no callout to Fig. 5E and Fig 5J in the text.

17)line 284: present two different mechanisms of ECR-targeted ..."

It would be helpful for the authors to briefly spell out exactly what they mean with these "two different mechanisms" to prevent confusion.

18)line 316: "targeting the TM"

Should TM read something else? E.g., TM domain?

19)line 327 and line 332:

Please indicate exactly which proteins and their constructs are being referred to, to prevent confusion and to work

towards Rigor and Reproducibility criteria.

20)line 336: 1 mM NiCl₂

The authors might consider mentioning in parentheses why NiCl₂ is added in the buffer.

21)line 398-402:

It is helpful in the Material and Methods to indicate what model was used to fit the SPR data, how many replicates were done, how many independent experiments were done, and if a standard deviation was calculated for the KD and how this was derived.

22)line 421-442

It is helpful in the Material and Methods to indicate what replicates and how many were taken along in the SRE-luciferase assays. How was the standard deviation derived, e.g., what is 'n' in n=3 in the legend for Fig. 2.

23)line 435-440

Please indicate how the proteins used for crystallization were purified.

24)Legend Fig. 1: LK27-31

Is this correct?

25)Fig. 1:

It would be helpful to indicate where T842G is in Fig. 1a.

26)Fig. 2

Please indicate in the legend that HBS is your buffer.

27)Fig. 3B

Given the high B-factor of the structure (104 Ang²) it would be very helpful for the authors to include a supplemental figure showing the quality of the electron density (2mFo-DFc contoured at 1sigma) for Fig. 3B, i.e. at the interface between LK30 and ADGRL3 lec.

28) Fig. 3C

This is a beautiful and very important figure. Please increase the size of this figure. It is very hard to see the green LK30 bound to the Lec domain surface. The authors might consider placing the conservation score bar vertically (or getting rid of it all together by displaying 100% conserved, semi-conserved, not conserved) to enable the figure to be enlarged. Please label key residues in Fig. 3C for LK30 (green stick model polypeptide shown)

29)Fig. 4A

Please label the ADGRL3 domains in the schematic (i.e., OLF, lec, etc.).

It would also be helpful to keep the same color scheme as Fig 1A if at all possible – perhaps the authors might consider this.

30)Fig. 5

What is n in the phrase "n=15 or n=10"

31)Table 1

The authors might consider mentioning why 29 TLS groups were used (is this higher than typically used?).

32)Fig. S1

Please include information on the replicates for Fig. S1C. Is a standard deviation available for the values in the Table S1C?

Please indicate what fractions were loaded on the SDS-Page gel for Fig. 4E so that the reader can see the correspondence with the chromatogram.

33) Please check references.

For example, information is missing in Ref. 5 and Ref. 29,

Reviewer #2 (Remarks to the Author):

The manuscript by Kordon et al. describes the identification and characterization of a synthetic antibody that specifically binds to the ECR domain of ADGRL3 and activates the receptor in an isoform-specific manner. The manuscript is well written, the article is technically sound and makes a valuable contribution to elucidating the mechanistic regulation of a GPCR. Overall, the conclusions drawn are supported by the experimental results, but there are some technical points that should be addressed:

1) SPR experiments:

the sensograms in Fig. 1C display the binding of LK30 to FL ECR and Lec/Olf domains immobilized onto a NTA sensor chip. In my experience, His-tag immobilization often suffers from a significant dissociation of the target protein from the chip, especially in multiple injection experiments. Have the authors experienced this problem and checked the amount of the immobilized ligand between different injections?

The two experiments (in fig 1C and D) are performed with the same concentrations of analyte, but the maximum RU in D is almost half than in C. How do the authors explain that? Is the amount of immobilized target similar in the two experiments?

Dissociation of the analyte does not appear to be complete, especially at higher concentrations, as expected in case of nanomolar affinities. Is there a regeneration step before the next injection?

Why the isolated Lec domain was not tested in SPR experiments?

In Fig S1C, the note says that the affinity of several sABs was determined by ELISA because SPR data could not be fit, but it is not clear which ones it refers to. Moreover, it is not described anywhere how these ELISA were performed (I found only the description of the single-point ELISA method in lines 376-380). Finally, the sensograms of the other experiments in the table (at least the ones that could be fit) should be shown in the supplementary information.

2) SEC experiments

In Fig. 1E, the Lec domain appears as a double band in the gel, while Lec-olf is a single band in S1E. Are these bands different glycosylation isoforms or is there degradation/proteolysis when using the isolated Lec domain?

In the SDS-PAGE of the SEC fractions in fig 4D-E, the bands corresponding to ADGRL3-ECR and LK30 have different apparent MW in the two gels (100 KDa and ~40 KDa in D and between 50-37 KDa and 25 KDa in E respectively). The band labeled as ADGRL3-ECR in E is almost the same size as that labeled as LK30 in D. How do the authors explain this?

3) Crystal structure:

The values of R_{merge} and R_{meas} seem to be unusually high in the higher resolution shell, although CC1/2 value could justify the resolution cutoff. It would be interesting to show an electron density map, maybe in Fig. 3B, surrounding the side chains of the residues involved in the interaction.

The structure solution and refinement methods are not easy to understand. The authors used 2 search probe models (6VHH and 4XWO) for molecular replacement, but they used the restrains from 5AFB and 5UCB for refinement. Is there a special reason for that? Why

not using the high-resolution structures also for MR? Why only rigid body with stereochemistry optimization and not individual coordinate refinement was performed in the last round?

In Table 1, there are 34 atoms for the ligands. Although I can understand the nature of the ligands from the deposition report, the reader has no idea what kind of ligand they are. This should be indicated somewhere in the text.

There are also only 7 solvent atoms in the model, with an average B factor lower than that of the proteins. I guess the authors only modeled few well-ordered water molecules in the structure. Are these located in the binding interface and/or mediate the interaction between AB and the lec domain?

4) SRE signaling

At lines 184-192: the authors state that LK30 activation depends on ADGRL3 autoproteolysis based on experiments with SRE-luciferase (fig. 2), while it binds to the T842G mutant with similar affinity as the wild type (and splicing variant) (fig. S2). However, according to the SPR experiments and the crystal structure, LK30 binds specifically to the Lec domain of the receptor, which is far away from the GPS in the GAIN domain (see Fig. 1A). How do they explain this effect? A possible explanation could be that autoproteolysis change the reciprocal orientation of the different domains of ECR. Is there any evidence of that? The authors should discuss these results in more detail.

Similarly, LK30 binds to ADGRL1 but does not activate it. On lines 294-297, the authors speculate that this may be due to differences in the sequence of the ECRs of the two aGPCRs. A graphical representation of the sequence homology between the ECRs of the two receptors (highlighting the most and least conserved regions) may help the reader.

In single-protein ELISA experiments the authors used fragments of human ADGRL3 and rat ADGRL1 (line 132) to evaluate the binding of selected antibodies. In addition, the experiments mentioned in Ref. 41 for the (SRE)-luciferase assay were performed with rat ADGRL1 and human ADGRL3. How similar is ADGRL1 in the two species? Since a single mutation in the autoproteolysis site abolished the LK30-induced signaling of ADGRL3, it should be clarified in the text which isoform (human or rat) of ADGRL1 was used in the other experiments described here (SRE-luciferase assay, flow cytometry) and, in case the rat isoform was used, if this does not affect the results.

At lines 169-173 the authors claim that LK30 acts as an agonist on SRE but not on cAMP signaling of ADGRL3. The activation of a specific downstream signaling pathway is remarkable and could be attributed to the shift in affinity of the LK30-activated receptor toward a specific G protein subtype. Are there other examples of partway-specific agonists for this class of receptors? Does the binding to extracellular physiological partners (e.g., TEN and FLRT) activate/inhibit the receptor? Although ADGRL3 signaling is described in some of the references provided, adding a brief description in the text would help the reader to better understand the system.

In ref 41, both rat Lphn1 and human Lphn3 display a similar basal activity in reducing the cAMP level in cells. Since LK30 also binds (likely) the Lec domain of Lphn1, it would be interesting to assess whether cAMP signaling is not affected in this isoform as well.

Minor points:

Fig 1F: Indicate the function used to fit the data. Error bars are visible for only some points. Are the others too low or not calculated?

Line 196: check reference format.

Reviewer #3 (Remarks to the Author):

Araç and coworkers isolated antibodies from a synthetic antibody library against the extracellular domains of GPCR subfamily members (ADGRL) that contain many independently folding extracellular domains. The authors state: "In this work, we employed the ADGRL subfamily of aGPCRs as a model system to demonstrate that the ECR of aGPCRs can be specifically targeted by antibodies." Would one expect anything else? These GPCRs contain several large extracellular domains and it can be regarded as almost certain that it will be possible to obtain antibodies against them.

Not unexpected, they obtained a set of antibodies and characterized two (LK30 and LK12) with a focus on LK30. This antibody as a scFv fragment, as shown by Xray crystallography, binds to a site in ADGRL3 which is also bound by a known ligand, TEN2. This interaction should be elucidated in more detail. What is the affinity of TEN2 binding to ADGRL3 and what is the IC50 of the LK30 antibody (to be determined e.g. by Biacore or BLI experiments)?

The authors showed that LK30 acts as an agonist, which is an interesting finding. However, the structural and functional reason for the agonistic behaviour remains unclear. The same holds true for the other very interesting finding, namely that the LK30 antibody is an agonist only for ADGRL3, but it does not activate the isoform ADGRL1, while affinity is in a similar range for both. This antibody and some others of the set may become excellent tool antibodies for functional discrimination between ADGRL3 and ADGRL1 but experiments are missing corroborating their value in functional assays in vitro or in vivo.

In conclusion, these are interesting data and the isolated antibodies are promising tools but from my perspective the data are too preliminary to be considered for the high impact journal Nature Communications.

Minor points:

Li 396 " ... functions 3927" citation numbering wrong

Figure 5: The concentration of the added antibody should be given.

Mat Meth: "The quality of purified sABs was analyzed by SDS-PAGE". These results should be shown in the supplement.

Response to Reviewer Comments

Isoform-and ligand-specific modulation of the adhesion GPCR ADGRL3/Latrophilin3 by a synthetic binder
NCOMMS-22-25191

Dear Dr. Savastano,

Thank you very much for the prompt handling of our manuscript. We are delighted that the reviewers appreciate the importance of our work for a variety of fields and thank them for their constructive comments. We have revised the manuscript accordingly. Below we provide our detailed responses:

Reviewer #1 (Remarks to the Author):

Comments to Authors:

This paper describes the development of a novel antibody that targets the lectin domain of ADGRL3. The figures are beautiful, and the experiments are well described.

We thank the reviewer for their appreciation of our work and enthusiasm.

There were some specific questions, however, that might benefit from some further clarification.

1) The authors write that antibodies targeting adhesion GPCRs have been developed before, i.e., Ref 22, Ref 28, and Ref 29. It is not clear how LK30 is different and a key advancement to the field. It would be helpful to spell this out for readers.

We thank the reviewer for these constructive comments. We agree that it is helpful to spell out the answer to these questions in the text. We modified the manuscript file to clarify these issues. Below we provide more detailed information about each point:

* is the binding site different/similar/same?

The antibodies reported in this study are the first that target the ADGRL subfamily of aGPCRs. Compared to other work, LK30 has been determined to bind to the N-terminal tip of the ADGRLs. It binds specifically to the Lectin domain, which is located the furthest from the transmembrane domain. Other previously reported antibodies, which were targeted to other aGPCRs bind to the GAIN domain that directly precedes the 7TM region or other domains (Bhudia et al 2020; Chatterjee et al. 2021; Salzman et al. 2017) or the precise epitope of the antibody was not defined (Bhudia et al 2020). The recent work by Stephan et al. 2022 reports an antibody targeting N-terminal PTX domain of ADGRD1/GPR133.

* is the affinity different/better/worse?

The affinity of the LK30 and other sABs reported in this work is in the low nanomolar range, which is comparable or better to values reported for the monobodies against ADGRG1/GPR56 (Chatterjee et al. 2021; Salzman et al. 2017). Affinity for other antibodies against aGPCR were not reported.

* are the other antibodies agonists or antagonists?

Most of the antibodies presented in previous work act as agonists, while two monobodies have been reported to be inverse agonists.

* is the selectivity different/similar/same towards ADGRs?

No other study has addressed the selectivity towards ADGRs. Our work is the first that aims to address whether it is possible to selectively target and act on different adhesion GPCRs. Thus, we used the

ADGRL/Latrophilin subfamily which are similar but different receptors. Our work is the first in the field to generate synthetic antibodies that can distinguish between similar isoforms. This is a very important property as it demonstrates the ability of antibodies to act specifically on aGPCR isoforms.

Our work -while addressing specificity- also provides observations about aGPCR activation that provide important insight into aGPCR activation mechanisms. For example, activation by other previously reported antibodies is independent of receptor cleavage (Chatterjee et al. 2021; Salzman et al. 2017), while LK30 activates ADGRL3 only when autoproteolysis occurs in the receptor.

2) The authors write that LK30 is unique because it is selective for ADGRL3 and inactive against ADGRL1. This finding is a bit puzzling.

We agree with the reviewer. We are also surprised by this result. However, the activation mechanism of adhesion GPCRs are still poorly understood and we expect that it involves more factors than previously thought. Please see our answer to Reviewer 2 point 4 for a full discussion of this issue.

LK30 binds ADGRL3 quite tightly, $KD \sim 4$ nM using SPR (i.e., biochemically). This reviewer could not find where it was stated how well LK30 binds ADGRL1 by SPR. LK30 binds ADGRL1 with $KD \sim 131$ nM and ADGLR1 ~ 147 nM, i.e. roughly similarly, by flow cytometry. Yet, LK30 is active against ADGRL3 (agonist) but not against ADGRL1 (inactive) in a cell-based assay (SRE-luciferase). LK30 is inactive against ADGRL3 using a cAMP based cell-assay, but this reviewer could not find whether LK30 is active against ADGRL1 in the cAMP based-assay.

To make a solid case that i) LK30 is selective (active against ADGRL3 and not ADGLR1), and ii) works allosterically (see Discussion), it would greatly strengthen the paper to fill in the missing information:

* show by SPR the binding affinity of LK30 for ADGRL1, as well as ADGRL3

We performed the suggested SPR experiments utilizing ECR fragments for both ADGRL1 and ADGRL3 as reported in Supplementary Figure 2 and Figure 1C,D,E, respectively. The binding affinity of LK30 to Lec domain of ADGRL3 is 7 nM and to Lec domain of ADGRL1 is 11 nM, confirming the strong interaction of LK30 with both receptors.

* make a sequence alignment comparing ADGRL3 and ADGRL1 in the region where ADGRL3 binds to LK30 (its CDR loops, Fig. 3b and Fig. 3c). Fig. 3c lacks a description of what sequences were compared so it is hard to assess. Also the conservation score in Fig. 3c is hard to assess. Which residues are identical? Which residues are semi-conserved and what residues were grouped together to form 'semi-conserved' designations?

We included a new Supplementary Figure 6 with sequence alignment of Lectin domains of both ADGRL1 and ADGRL3. We did a detailed analysis of the identical and semi-conserved residues and highlighted areas/residues responsible for LK30 binding. We mapped this information on the Lec domain structure (as suggested by reviewer in point 10 here). Additionally, in Supplementary Figure 6, we included a graphical representation of the sequence homology between domains of ADGRL3 and ADGRL1, as requested by reviewer 2.

* explain why LK30 is active ADGRL3 in a SRE-luciferase assay (but ADGRL1 is not); while it is not active against ADGRL3 in the cAMP based assay, and state whether LK30 works against ADGRL1 in the cAMP based assay.

We hypothesize that LK30 can act as a biased agonist for ADGRL3 in the SRE assay (G12/13 pathway), as it does not influence receptor activity in the cAMP assay. We also performed additional experiments showing that LK30 is a neutral binder for ADGRL1 in both the SRE assay (Figure 2) and the cAMP-based assay (Supplementary Figure 4).

However, the experimentation needed for a comprehensive explanation of this bias would require additional extensive structural and functional analysis utilizing full-length receptor, which is beyond the scope of this current study.

3) line 129: protein L affinity,
Please define

Protein L affinity chromatography is a standard procedure used for the purification of antibody and antibody fragments that contain kappa light chains (in contrast to Protein A and Protein G, which bind to the Fc region of immunoglobulins).

4) line 134: while one sAB recognized the Olf domain
Which sAB? Experiment shows that Olf domain or both Olf-Lec domains are recognized, correct? If so, please revise

The reviewer is correct that the construct containing both Lec and Olf domains was recognized by sAB LK12. However, since LK12 did not bind to the isolated Lec domain, we suspect the Olf domain is being recognized by LK12. As this interpretation is not fully supported by other experiments (e.g. utilizing isolated Olf domain for binding experiments), and it is possible that the antibody can bind to the epitope coming from combined domains, we revised the sentence accordingly.

5) line 159: SRE-luciferase assay
It is very helpful for the reader to quickly explain how this assay works to assess ADGRL1/3 activity.

We added a short explanation of the assay in the main text.

6) line 165-168: despite ability of ADGRL1 to bind LK30 (Fig. 2A, B)
On what basis does ADGRL1 bind to LK30? See also point 2).

We had previously used flow cytometry assays and showed that LK30 binds to ADGRL1 expressed on the cell surface (Figure 1F). As suggested by the reviewer, we also performed SPR analysis of LK30 binding to the purified ADGRL1 ECR fragments (new Supplementary Figure 2).

7) line 169-170: cAMP-based assay
It is very helpful for the reader to quickly explain how this assay works to assess ADGRL1/3 activity.

We added a short explanation of the assay in the main text.

8) line 172-173: LK30 has no effect on ADGR3 in cAMP assay
Please explain why this might be. Please indicate if LK30 is active against ADGRL1 in the cAMP assay.

LK30 might act as a biased agonist towards one of the signaling pathways of ADGRL3. It had been previously shown that ADGRL3 signals through G12/13, Gi and Gq. Our SRE results suggest that LK30 binding induces ADGRL3 signaling through G12/13 (which can activate the SRE); however, since there might be other proteins involved in the SRE signaling pathway, we did not want to overinterpret our results, by distinctly stating the G12/13-bias.

As reviewers suggested, we performed the cAMP assay utilizing both ADGRL3 and ADGRL1 and did not observe LK30 affecting the signaling of any of the receptors (updated Supplementary Figure 4).

9) line 176: recent studies and models for autoprolytic mechanism
Perhaps, the authors would like to revise lines 176-182 to include recent studies, e.g., the recent four studies summarized in Boucard (2022) Nature Vol 604, p. 628

Thank you for this suggestion, we revised this paragraph and included the newest aGPCR structure references.

10)line 218-219: conserved region on N-term part of Lec

Please indicate in the Material and Methods and in the legend for Fig. 3c, which sequences were used for the multi-sequence alignment. Please indicate if sequences for ADGRL3 and ADGRL1 both were compared or only for ADGRL3.

The appropriate explanation in the Methods section has been included:

For the conservation analysis, the ConSurf server was utilized using the software default parameters. 150 sequences that sample the list of homologs to reference (human ADGRL3_{Lec}) was searched using HMMER search algorithm in UNIREF-90 database. Sequences with minimum of 35% and maximum 95% identity were used. As a result, these sequences were a mixture of all ADGRL isoforms. After aligning with MAFFT-L-INS-i method, the conservation was calculated using Bayesian method and mapped to the surface of ADGRL3_{Lec} in UCSF Chimera.

Please add a version of the figure where the sequence conservation between ADGRL3 and ADGRL1 in the Lec domain is shown. Please show which residues are 100% identical and which are semi-conserved. Please also indicate how residues are grouped for the semi-conserved analysis. Please include the multi-sequence alignment as a supplemental figure.

The sequence alignment between ADGRL1 and ADGRL3 was included in the Supplementary Figure 6, with proper presentation of the residues that are identical and semi-conserved. The sequence conservation between Lectin domains of ADGRL3 and ADGRL1 was also mapped on the ADGRL3_{Lec} structure as suggested.

11)line 227-228: D67 and S38

Please indicate in Fig. 4C (with little arrows) where D67 and S38 are. Please indicate in Fig. 3B (label) where S38 is.

Both Figure 4C and 3B has been updated accordingly.

12)line 229: buried in hydrophobic pocket in the LK30

Please indicate “.....buried in hydrophobic pocket of ??whom??. in the LK30.....”

The proper clarification was included.

13) line 242-244

It is helpful to explain to the reader why disruption of the ADGRL3-teneurin complex results in increased activity in the SRE-luciferase assay.

Here we are talking about two separate events: 1). Disruption of ADGRL3/TEN2 complex, and 2). Increase of basal signaling of ADGRL3, which is in the absence of teneurin and is unrelated to the presence or absence of teneurin (-although we cannot exclude the possibility of an effect from the endogenous teneurin that may exist in mammalian HEK293T cells we use in our assay.) Both events require LK30 binding to ADGRL3 to occur, but at this point we cannot correlate them together. We were not able to test the effect of TEN2 interaction on ADGRL3-mediated SRE signaling. However, previous reports (Li et al. 2018) suggested that TEN2 binding to ADGRL3 causes a decrease in cAMP signaling assays. Thus, LK30 could potentially act as a molecular switch of the ADGRL3 signaling by disrupting the TEN2/ADGRL3 complex and inducing SRE-mediated response, More experiments would be needed to confirm this concept. Thus, we are not claiming this in our paper.

14)line 259: “of ADGRL3/FLRT3 interaction”

Should this read “on ADGRL3/FLRT3 interaction....”?

The reviewer is correct, the typo has been corrected.

15)line 261: ADGRL3 Olf domain binder

Shouldn't this be Olf domain or Olf-lec tandem? Sometimes antibodies can recognize epitopes coming from different domains.

As in point 4), we have revised the sentence to the Lec/Olf binder to avoid confusion.

16)There is no callout to Fig. 5E and Fig 5J in the text.

We have added callouts to Figures 5E and J in proper parts of the text.

17)line 284: present two different mechanisms of ECR-targeted ..."

It would be helpful for the authors to briefly spell out exactly what they mean with these "two different mechanisms" to prevent confusion.

We added a brief explanation of the two mechanisms of activation mentioned in the paragraph.

18)line 316: "targeting the TM"

Should TM read something else? E.g., TM domain?

The reviewer is right, the typo was corrected to 7TM (as introduced in the text previously as seven transmembrane helix region).

19)line 327 and line 332:

Please indicate exactly which proteins and their constructs are being referred to, to prevent confusion and to work towards Rigor and Reproducibility criteria.

The detailed information was included in the method section.

20)line 336: 1 mM NiCl₂

The authors might consider mentioning in parentheses why NiCl₂ is added in the buffer.

The protocol used for secreted proteins purification is a standard protocol used in the lab (see Leon et al 2020, Li et al 2020, etc.). Addition of nickel chloride during the first incubation step is to minimize binding of non-specific contaminants present in the media during the Ni-NTA incubation step.

21)line 398-402:

It is helpful in the Material and Methods to indicate what model was used to fit the SPR data, how many replicates were done, how many independent experiments were done, and if a standard deviation was calculated for the KD and how this was derived.

In the case where SPR is used only to screen antibodies and to identify their epitope, we typically only performed one kinetic experiment as in our experience, these results are highly reproducible. In the case of LK30, we performed the experiments in duplicates. The K_D values obtained for binding of LK30 to different fragments of ADGRL3 are very similar, further confirming the reproducibility of the experiment. These data were fit using Langmuir model (1:1 binding) with drifting baseline. This information is now included in the Material and Methods section.

22)line 421-442

It is helpful in the Material and Methods to indicate what replicates and how many were taken along in the SRE-luciferase assays. How was the standard deviation derived, e.g., what is 'n' in n=3 in the legend for Fig. 2.

We included a short explanation of data analysis for both SRE and cAMP signaling assays in the Methods section.

23)line 435-440

Please indicate how the proteins used for crystallization were purified.

The proteins used for X-ray crystallography were purified as described under “Protein expression and purification”, “Cloning, Overexpression and Purification of sABs” and “Formation of ADGRL3/LK30 Complex” sections. To avoid unnecessary redundancy, authors did not want to repeat the same protocols twice.

24) Legend Fig. 1: LK27-31

Is this correct?

The improper legend was corrected.

25) Fig. 1:

It would be helpful to indicate where T842G is in Fig. 1a.

To keep all figures as clear as possible, instead of updating Figure 1A, we decided to include an additional panel in Supplementary Figure 3, that includes the location of the mutation site within the GPS motif.

26) Fig. 2

Please indicate in the legend that HBS is your buffer.

The figure legend includes the buffer info now.

27) Fig. 3B

Given the high B-factor of the structure (104 Å) it would be very helpful for the authors to include a supplemental figure showing the quality of the electron density (2mF_o-DF_c contoured at 1σ) for Fig. 3B, i.e. at the interface between LK30 and ADGRL3 lec.

The updated Supplementary Figure 6 has been added to the manuscript to present the quality of the electron density. The high B-factor may be due to the known “elbow problem” with sAB structures. (<https://www.ncbi.nlm.nih.gov/pmc/articles/PMC5800945/>)

28) Fig. 3C

This is a beautiful and very important figure. Please increase the size of this figure. It is very hard to see the green LK30 bound to the Lec domain surface. The authors might consider placing the conservation score bar vertically (or getting rid of it all together by displaying 100% conserved, semi-conserved, not conserved) to enable the figure to be enlarged. Please label key residues in Fig. 3C for LK30 (green stick model polypeptide shown)

Based on this suggestion, we decided to increase the size of panel C as much as possible. As discussed in points 2 and 10 here, the new Supplementary Figure 7 has been included in the manuscript to address the additional issues with conservation analysis.

29) Fig. 4A

Please label the ADGRL3 domains in the schematic (i.e., OLF, lec, etc.). It would also be helpful to keep the same color scheme as Fig 1A if at all possible – perhaps the authors might consider this.

Agreed and done.

In Figure 4A, we used a slightly different coloring scheme here to avoid including too many colors in one figure, as it might confuse the reader. Instead, each of the discussed proteins have been assigned its color (pink for TEN2, blue for FLRT3, yellow for ADGRL3 and green for sAB to be consistent with colors presented in the structure in Figure 3), and only the domains taking part in complex formation have been highlighted.

30) Fig. 5

What is n in the phrase “n=15 or n=10”

Updated legend description has been included.

Quantification of aggregation index are presented as mean \pm SD from either 15 or 10 images ($n = 15$ or $n = 10$ for LK12 experiments) collected in three independent experiments, **** $p < 0.0001$; one-way ANOVA.

31)Table 1

The authors might consider mentioning why 29 TLS groups were used (is this higher than typically used?).

TLS groups were automatically defined in Phenix.refine. The relatively high number is potentially due to the sAB dynamics in the elbow region as previously stated.

32)Fig. S1

Please include information on the replicates for Fig. S1C. Is a standard deviation available for the values in the Table S1C?

The information about replicates is now included in the Methods section. Since in most cases only individual SPR measurement was performed, there are no SD values calculated.

Please indicate what fractions were loaded on the SDS-Page gel for Fig. 4E so that the reader can see the correspondence with the chromatogram.

The SEC fractions loaded on the gel are now marked with the gray area on the chromatograms.

33)Please check references.

For example, information is missing in Ref. 5 and Ref. 29,

The references have been carefully inspected and missing information have been added if possible.

We thank the reviewer for these constructive comments. We believe the manuscript is now in a much better shape after addressing these comments.

Reviewer #2 (Remarks to the Author):

The manuscript by Kordon et al. describes the identification and characterization of a synthetic antibody that specifically binds to the ECR domain of ADGRL3 and activates the receptor in an isoform-specific manner. The manuscript is well written, the article is technically sound and makes a valuable contribution to elucidating the mechanistic regulation of a GPCR. Overall, the conclusions drawn are supported by the experimental results, but there are some technical points that should be addressed:

We thank the reviewer for their appreciation of our work and address their comments below,.

1) SPR experiments:

the sensograms in Fig.1C display the binding of LK30 to FL ECR and Lec/Olf domains immobilized onto a NTA sensor chip. In my experience, His-tag immobilization often suffers from a significant dissociation of the target protein from the chip, especially in multiple injection experiments. Have the authors experienced this problem and checked the amount of the immobilized ligand between different injections?

Indeed, we occasionally observe dissociation on the Ni-NTA chips. For that reason, we always test each target before proceeding with the kinetic experiment. In some cases, we extend His-tag from standard 6xHis to 10xHis or change the immobilization method entirely. In the case of ADGRLs, after immobilization, we did not observed dissociation from the chip for several minutes. Additionally, before

each round of antibody binding, we regenerate the chip with EDTA and immobilize fresh portion of the target protein.

The two experiments (in fig 1C and D) are performed with the same concentrations of analyte, but the maximum RU in D is almost half than in C. How do the authors explain that? Is the amount of immobilized target similar in the two experiments?

Yes, in both cases we immobilized same amount of protein. However, the system we are using allows for analysis of multiple analytes simultaneously. For that reason, not all the sABs were tested in the same channels on the chips. Despite using same amount of ligand RU varies between some channels resulting in different RU for the same concentration of the sAB.

Dissociation of the analyte does not appear to be complete, especially at higher concentrations, as expected in case of nanomolar affinities. Is there a regeneration step before the next injection?

Yes, as mentioned above, we do regenerate the chip and immobilize fresh amount of protein between each injection.

Why the isolated Lec domain was not tested in SPR experiments?

Initially, the SPR experiments were performed using only constructs chosen from the phage display selections. However, based on the reviewer's suggestion, we performed additional SPR experiments utilizing separated Lec domains of both ADGRL3 and ADGRL1 to further confirm the LK30 binding to Lec domains of both receptors. The new data can be found in the updated Figure 1 and Supplementary Figure 2.

In Fig S1C, the note says that the affinity of several sABs was determined by ELISA because SPR data could not be fit, but it is not clear which ones it refers to. Moreover, it is not described anywhere how these ELISA were performed (I found only the description of the single-point ELISA method in lines 376-380). Finally, the sensograms of the other experiments in the table (at least the ones that could be fit) should be shown in the supplementary information.

We have specified the affinity information and added the requested method section and sensograms to the Supplementary Figure 1.

2) SEC experiments

In Fig. 1E, the Lec domain appears as a double band in the gel, while Lec-olf is a single band in S1E. Are these bands different glycosylation isoforms or is there degradation/proteolysis when using the isolated Lec domain?

The extracellular domains used in our work were expressed and purified from insect cells which allows for posttranslational modifications of proteins, including glycosylation. The double band associated with the lectin domain is due to different glycosylation patterns – when present on a small 11 kDa Lectin domain, they can be separate on the SDS-PAGE 15% gel. With the larger 45 kDa Lec/Olf constructs, we believe the size differential is not enough to separate them on SDS-PAGE using our standard running protocol.

In the SDS-PAGE of the SEC fractions in fig4D-E, the bands corresponding to ADGRL3-ECR and LK30 have different apparent MW in the two gels (100 KDa and ~40 KDa in D and between 50-37 KDa and 25 KDa in E respectively). The band labeled as ADGRL3-ECR in E is almost the same size as that labeled as LK30 in D. How do the authors explain this?

In Figure 4D, the full-length ECR of Lphn3 was used (consisting of Lec-Olf-HormR-GAIN domains), which has the molecular weight of 97kDa, due to the need of co-expression of TEN2/LPHN3 constructs. In

Figure 4E, the shorter, Lec/Olf constructs were used (~42kDa). We updated labels in the Figure 4 with more detailed constructs

The difference in sAB size is due to the reducing agent present in the sample buffer. In D, the disulfide bonds between heavy and light chains of the sAB did not get fully reduced (potentially due to the low concentration of reducing agent or shorter incubation with the buffer), which made the sAB migrate on the gel as a single ~50kDa band. In E (and Fig. 1F), disulfide bonds were properly disrupted by reducing agent, disrupting the heavy chain:light chain interaction, which is visualized by the collapsed ~25kDa band (size of the separated heavy/light chain). For comparison, please refer to Supplementary Figure 1C for the SDS-PAGE analysis of purified sABs performed in either reducing/non-reducing conditions.

3) Crystal structure:

The values of R_{merg} and R_{meas} seem to be unusually high in the higher resolution shell, although CC1/2 value could justify the resolution cutoff. It would be interesting to show an electron density map, maybe in Fig. 3B, surrounding the side chains of the residues involved in the interaction.

We report statistics for both the entire collected dataset (Table 1, data collection statistics) and for the data used in refinement (refinement statistics). We cut the data for refinement using CC1/2. The new Supplementary Figure 6 has been added to the manuscript to present the quality of the electron density.

The structure solution and refinement methods are not easy to understand. The authors used 2 search probe models (6VHH and 4XWO) for molecular replacement, but they used the restraints from 5AFB and 5UCB for refinement. Is there a special reason for that? Why not using the high-resolution structures also for MR? Why only rigid body with stereochemistry optimization and not individual coordinate refinement was performed in the last round?

Initially we used known structures for molecular replacement, and we did not think to search for higher resolution structures. However, using the original search models for MR followed by manual building, our refinements reached a plateau. We may have gotten stuck in a “local minimum” of the refinement process. Using restraints from these high-resolution structures (5AFB and 5UCB) allowed us to continue in our refinements. We chose to continue this way because we had already manually built much of our model.

The quality of the density in some regions of the structure was not as high compared to the interface area, potentially due to the sAB dynamic elbow region (see Bailey, L. J. *et al.* Locking the Elbow: Improved Antibody Fab Fragments as Chaperones for Structure Determination. *J. Mol. Biol.* 2018 430, 337–347). In the final rounds of refinement, we found that allowing individual coordinate refinement introduced many geometric and stereochemical errors at the elbow region which could only be fixed manually. We fixed these errors manually and continued with rigid body refinement.

In Table 1, there are 34 atoms for the ligands. Although I can understand the nature of the ligands from the deposition report, the reader has no idea what kind of ligand they are. This should be indicated somewhere in the text.

We have added an updated explanation in the results section. Ligands in the structure include glycerol and PEG, both present in the crystallization solution or cryoprotectant, and are assigned due to their shape. We assigned several large densities as sulfates as their densities are too big to be water molecules and they interact with positively charged residues. Sulfate is present in the crystallization solution.

There are also only 7 solvent atoms in the model, with an average B factor lower than that of the proteins. I guess the authors only modeled few well-ordered water molecules in the structure. Are these located in the binding interface and/or mediate the interaction between AB and the lec domain?

At this resolution, we were only confident modeling the most well-ordered waters. There is one water near the binding interface in one copy of the structure (water 16; near Chain E, E67), but the majority of the interface is not mediated by water.

4) SRE signaling.

At lines 184-192: the authors state that LK30 activation depends on ADGRL3 autoproteolysis based on experiments with SRE-luciferase (fig. 2), while it binds to the T842G mutant with similar affinity as the wild type (and splicing variant) (fig. S2). However, according to the SPR experiments and the crystal structure, LK30 binds specifically to the Lec domain of the receptor, which is far away from the GPS in the GAIN domain (see Fig. 1A). How do they explain this effect? A possible explanation could be that autoproteolysis change the reciprocal orientation of the different domains of ECR. Is there any evidence of that? The authors should discuss these results in more detail.

All reviewers have asked the question: How does the antibody activate the adhesion GPCR? This can be split into two subquestions: 1). Why does LK30 activation depend on the ADGRL3 autoproteolysis? 2). Why does LK30 activate only ADGRL3 and not ADGRL1 although it binds to both receptors with similar binding affinities? The simple answer is that we do not know. In our study, we want to report these observations because we believe they are important observations.

In summary, the adhesion GPCR field currently appears to favor the “Stachel peptide mediated activation mechanism”. A number of papers talk about this mechanism and the numerous transmembrane structures bound to the Stachel peptide. We generally support this mechanism. However, we also think that this is too simplistic and that this mechanism does NOT explain numerous observations some of which are discussed in our Salzman, 2018, PNAS paper and some that are reported by other groups. Very briefly, Stachel-peptide mechanism does not explain how aGPCRs that are not cleaved are activated, or how some in vivo phenotypes does not depend on autoproteolysis, or how the receptor can be turned off. The observations we report in this manuscript add to the other “puzzling” results. We very much want to understand how exactly adhesion GPCRs work. For this reason, we are pursuing experiments that span cryo EM, single molecule FRET, and biochemistry; and we plan to report our ideas in a separate manuscript when it is ready. We prefer not to discuss the mechanism too much in this manuscript since clarifying data should be forthcoming.

Still, to answer the reviewers' questions as much as possible under the limited time, we performed additional experiments to monitor whether antibody leads to receptor activation by causing separation of the ECR from the transmembrane anchored part to reveal the Stachel peptide (see Supplementary Fig. 5). Our results suggest this is not the mechanism.

Similarly, LK30 binds to ADGRL1 but does not activate it. On lines 294-297, the authors speculate that this may be due to differences in the sequence of the ECRs of the two aGPCRs. A graphical representation of the sequence homology between the ECRs of the two receptors (highlighting the most and least conserved regions) may help the reader.

We added a ADGRL scheme with sequence identity between discussed receptors in new Supplementary Figure 7.

In single-protein ELISA experiments the authors used fragments of human ADGRL3 and rat ADGRL1 (line 132) to evaluate the binding of selected antibodies. In addition, the experiments mentioned in Ref. 41 for the (SRE)-luciferase assay were performed with rat ADGRL1 and human ADGRL3. How similar is ADGRL1 in the two species? Since a single mutation in the autoproteolysis site abolished the LK30-induced signaling of ADGRL3, it should be clarified in the text which isoform (human or rat) of ADGRL1 was used in the other experiments described here (SRE-luciferase assay, flow cytometry) and, in case the rat isoform was used, if this does not affect the results.

All further experiments were performed utilizing human ADGRL3 and rat ADGRL1. Since the rat isoform of ADGRL1 shows 98% identity to the human ADGRL1, we do not expect it to affect the results of our work.

At lines 169-173 the authors claim that LK30 acts as an agonist on SRE but not on cAMP signaling of ADGRL3. The activation of a specific downstream signaling pathway is remarkable and could be attributed to the shift in affinity of the LK30-activated receptor toward a specific G protein subtype. Are there other examples of partway-specific agonists for this class of receptors? Does the binding to extracellular physiological partners (e.g., TEN and FLRT) activate/inhibit the receptor? Although ADGRL3 signaling is described in some of the references provided, adding a brief description in the text would help the reader to better understand the system.

There are no other examples of pathway-specific agonists for ADGRLs. These are the first antibodies engineered against ADGRLs and no published small molecules exist for ADGRLs. We added a brief description in the text.

In ref 41, both rat Lphn1 and human Lphn3 display a similar basal activity in reducing the cAMP level in cells. Since LK30 also binds (likely) the Lec domain of Lphn1, it would be interesting to assess whether cAMP signaling is not affected in this isoform as well.

We repeated cAMP signaling assay to include both ADGRL1 and ADGRL3, and show the cAMP signaling is not affected by any of the isoforms (See updated Supplementary Figure 4).

Minor points:

Fig 1F: Indicate the function used to fit the data. Error bars are visible for only some points. Are the others too low or not calculated?

The concentration-response curve in GraphPad Prism was used to fit the data and it is now reported in the figure legend. Errors for all data points have been calculated – some error bars were too low to properly display on the diagram.

Line 196: check reference format.

The reference has been corrected.

We thank the reviewer for these constructive comments. We believe the manuscript is now in a much better shape after addressing these comments.

Reviewer #3 (Remarks to the Author):

Araç and coworkers isolated antibodies from a synthetic antibody library against the extracellular domains of GPCR subfamily members (ADGRL) that contain many independently folding extracellular domains. The authors state: “In this work, we employed the ADGRL subfamily of aGPCRs as a model system to demonstrate that the ECR of aGPCRs can be specifically targeted by antibodies.” Would one expect anything else? These GPCRs contain several large extracellular domains and it can be regarded as almost certain that it will be possible to obtain antibodies against them.

We thank the reviewer for their comments.

The reviewer is correct that obtaining antibodies and/or antibody fragments against extracellular domains of other adhesion receptors is possible. However, utilizing the phage display library we aimed to generate ADGRL-specific sABs that do not only act only as a simple binder for immunochemistry assays, but also

be utilized as research tools for aGPCR signaling studies and/or for elucidation of their interaction with endogenous binding partners.

Not unexpected, they obtained a set of antibodies and characterized two (LK30 and LK12) with a focus on LK30. This antibody as a scFv fragment, as shown by Xray crystallography, binds to a site in ADGRL3 which is also bound by a known ligand, TEN2. This interaction should be elucidated in more detail. What is the affinity of TEN2 binding to ADGRL3 and what is the IC50 of the LK30 antibody (to be determined e.g. by Biacore or BLI experiments)?

The reviewer makes a good point about expanding the TEN2/ADGRL3 binding experiments. We determined the affinity of ADGRL3 binding to TEN2 using flow cytometry experiments (Please refer to the new Supplementary Figure 8).

Briefly, FL TEN2 was expressed on HEK293T cell surface, then purified and biotinylated Lec domain of ADGRL3 was added in increasing concentrations to the cell sample. Then, Lec domain binding to cell-surface expressed teneurin was detected by flow cytometry by addition of fluorescently labeled neutravidin, and binding affinity was calculated by fitting the concentration-response curve in GraphPad Prism (EC50=130 nM) (Supplementary Figure 8A). We also performed a competition experiment in which we used a saturating concentration (5 μ M) of Lec domain to bind to the TEN2 expressed on the cell surface. We then added the LK30 antibody at increasing concentrations to observe the dissociation of the Lectin domain from teneurin because the antibody binds to teneurin at the same binding site as the Lectin domain. We measured IC50 as 190 nM (Supplementary Figure 8B).

The authors showed that LK30 acts as an agonist, which is an interesting finding. However, the structural and functional reason for the agonistic behaviour remains unclear. The same holds true for the other very interesting finding, namely that the LK30 antibody is an agonist only for ADGRL3, but it does not activate the isoform ADGRL1, while affinity is in a similar range for both. This antibody and some others of the set may become excellent tool antibodies for functional discrimination between ADGRL3 and ADGRL1 but experiments are missing corroborating their value in functional assays *in vitro* or *in vivo*.

We agree about the potential of the antibodies as excellent tools and that more detailed functional *in vivo* assays would be of great interest. However, this is outside the scope of this study and we focused on the generation and biochemical/structural characterization and analysis of the antibody fragments utilizing purified proteins and *in vitro* signaling and cell aggregation assays.

In conclusion, these are interesting data and the isolated antibodies are promising tools but from my perspective the data are too preliminary to be considered for the high impact journal Nature Communications.

Minor points:

Li 396 " ... functions 3927" citation numbering wrong

The reference has been corrected.

Figure 5: The concentration of the added antibody should be given.

The concentration of the sAB (5 μ M) was mentioned in the figure legend.

Mat Meth: "The quality of purified sABs was analyzed by SDS-PAGE". These results should be shown in the supplement.

The SDS-PAGE analysis of all mentioned sABs is now included in the Supplementary Figure 1.

We thank all the reviewers for their constructive comments.

REVIEWERS' COMMENTS

Reviewer #2 (Remarks to the Author):

The authors have successfully addressed all my requests. I believe that the manuscript is now suitable for publication.

Reviewer #3 (Remarks to the Author):

This is a revision of a previously submitted manuscript which I rejected for publication due to missing experimental data. I am pleased to see that the authors performed a series of additional experiments that now shed some light on the properties and functions of the adhesion GPCR-specific antibodies that were isolated during this work. My critical points were all addressed as were, as far as I can oversee this, the points raised by the other two reviewers. In conclusion, I can recommend the manuscript now for publication.

Response to Reviewer Comments

Isoform-and ligand-specific modulation of the adhesion GPCR ADGRL3/Latrophilin3 by a synthetic binder

NCOMMS-22-25191A

Reviewer #2 (Remarks to the Author):

The authors have successfully addressed all my requests. I believe that the manuscript is now suitable for publication.

We thank the reviewer for their appreciation of our work and recommending our manuscript for publication.

Reviewer #3 (Remarks to the Author):

This is a revision of a previously submitted manuscript which I rejected for publication due to missing experimental data. I am pleased to see that the authors performed a series of additional experiments that now shed some light on the properties and functions of the adhesion GPCR-specific antibodies that were isolated during this work. My critical points were all addressed as were, as far as I can oversee this, the points raised by the other two reviewers. In conclusion, I can recommend the manuscript now for publication.

We thank the reviewer for their comments and suggestions, and we are happy they now recommend our revised work for publication.